



Technical Note: Partial wavelet coherency for improved understanding of
scale-specific and localized bivariate relationships in geosciences
Wei Hu[1] and Bing Si[2]
*[1]The New Zealand Institute for Plant and Food Research Limited, Private Bag 4704, Christchurch 8140,*
*New Zealand*
*[2]University of Saskatchewan, Department of Soil Science, Saskatoon, SK S7N 5A8, Canada*
*Correspondence to:* Wei Hu (wei.hu@plantandfood.co.nz)
**Abstract**
Bivariate wavelet coherency is widely used to untangle the scale-specific and localized
bivariate relationships in geosciences. However, it is well-known that bivariate
relationships can be misleading when both variables are correlated to other variables. Partial
wavelet coherency (PWC) has been proposed, but is limited to one excluding variable and
presents no phase information. We aim to develop a new PWC method that can deal with
multiple excluding variables and presents phase information for the PWC. Tests with both
stationary and non-stationary artificial datasets verified the known scale- and localized
bivariate relationships after eliminating the effects of other variables. Compared with the
previous PWC method, the new method has the advantages of capturing phase information,
dealing with multiple excluding variables, and producing more accurate results. The new
method was also applied to two field measured datasets. Results showed that the coherency





between response and predictor variables was usually less affected by excluding variables
when predictor variables had higher correlation with the response variable. Application of
the new method also confirmed the best predictor variables for explaining temporal
variations in free water evaporation at Changwu site in China and spatial variations in soil
water content in a hummocky landscape in Saskatchewan Canada. We suggest the PWC
method to be used in combination with previous wavelet methods to untangle the scale-
specific and localized multivariate relationships in geosciences. The PWC calculations
were coded with Matlab and are available in the supplement.

## 1. Introduction

Geoscience data, such as spatial distribution of soil moisture in undulating terrains and
temporal series of climatic variables, usually consist of a variety of transient processes with
different frequencies (scales) that may be localized in time or space (Graf et al., 2014; Si,
2008; Torrence and Compo, 1998). Wavelet methods are widely used to detect scale-
specific and localized features of geoscience data irrespective of whether they are stationary
or non-stationary. Among which, bivariate wavelet coherency (BWC) is widely accepted as
a tool for detecting scale-specific and localized bivariate relationships in a range of areas in
geoscience (Biswas and Si, 2011a; Das and Mohanty, 2008; Lakshmi et al., 2004; Polansky
et al., 2010; Si and Zeleke, 2005). Recently, Hu and Si (2016) have extended the BWC to
multiple wavelet coherence (MWC) that can be used to untangle multivariate (≥3 variables)
relationships in multiple scale-location domains. This method has been successfully used
in hydrology (Gu et al., 2020; Hu et al., 2017; Mares et al., 2020; Nalley et al., 2019; Su et



al., 2019) and other areas such as soil science (Centeno et al., 2020), environmental science
(Zhao et al., 2018), climate (Song et al., 2020), and economics (Sen et al., 2019).

The MWC application has shown that an increased number of predictor variables does

not necessarily explain more variation in the response variable, partly because predictor
variables are usually cross-correlated (Hu and Si, 2016). For the same reason, bivariate
relationships can be misleading if the predictor variable is correlated with other variables
that control the response variable. Partial correlation analysis is one such method to deal
with this issue (Kenney and Keeping, 1939), but the extension of partial correlation to the
multiple scale-location domain is limited. In order to better understand the bivariate
relationships at multiple scales and locations, the BWC needs to be extended to partial
wavelet coherency (PWC) by eliminating the effects of other variables.

The BWC was extended to PWC by Mihanović et al. (2009). Their method has been

widely employed in the areas of marine science (Ng and Chan, 2012a, b), climate
(Rathinasamy et al., 2017; Tan et al., 2016), greenhouse gas emissions (Jia et al., 2018; Li
et al., 2018; Mutascu and Sokic, 2020), and economics (Aloui et al., 2018; Altarturi et al.,
2018; Wu et al., 2020), among others. However, Mihanović et al. (2009) considered one
excluding variable only and did not include the phase angle difference between response
and predictor variables.

As an extension of previous studies (Hu and Si, 2016; Mihanović et al., 2009), this paper

aims to develop a PWC method that considers more than one excluding variable and phase
information. The new method is developed in analogy with the partial coherency in the





multiple spectral case (Koopmans, 1995). It is first tested with artificial datasets following
Yan and Gao (2007) and Hu and Si (2016) to demonstrate its capability of capturing the
known relationships of the artificial data. Next, the new method is compared with the
Mihanović et al. (2009) method. Then it is applied to two real (i.e., field measured) datasets
in geosciences including temporal series of free water evaporation at the Changwu site in
China (Hu and Si, 2016) and spatial series of soil water content from a transect in the
hummocky landscape in Saskatchewan, Canada (Biswas and Si, 2011a; Hu et al., 2017).
These two datasets are chosen because the MWC results previously presented (Hu and Si,
2016; Hu et al., 2017) can be used to assess the new method.
**2. Theory**
Similar to BWC and MWC, PWC is calculated from auto- and cross-wavelet power
spectra, for the response variable $y$, predictor variable $x$, and excluding variables $Z$ ($Z =$
$\{Z_1, Z_2, \cdots, Z_q\}$). In analogy with the partial coherency in the multivariate spectral case
(Koopmans, 1995), the complex PWC between $y$  and $x$  after excluding variables $Z$ at
scale $s$ and location $\tau$, $\gamma_{y,x \cdot Z}$, can be written as:
$$\gamma_{y,x \cdot Z}(s,\tau) = \frac{\left(1 - R^2_{y,x \cdot Z}(s,\tau)\right)\gamma_{y,x}(s,\tau)}{\left(\left(1 - R^2_{y,Z}(s,\tau)\right)\left(1 - R^2_{x,Z}(s,\tau)\right)\right)^{1/2}} \qquad (1)$$
where $R^2_{y,x \cdot Z}(s,\tau)$, $R^2_{y,Z}(s,\tau)$, and $R^2_{x,Z}(s,\tau)$  can be calculated by following Hu and Si
(2016) as





$\qquad R_{y,x \cdot Z}^2(s,\tau) = \dfrac{\underset{W}{\leftrightarrow}^{y,Z}(s,\tau)\, \underset{W}{\leftrightarrow}^{Z,Z}(s,\tau)^{-1}\, \overline{\underset{W}{\leftrightarrow}^{x,Z}(s,\tau)}}{\underset{W}{\leftrightarrow}^{y,x}(s,\tau)}$ (2)
$\qquad R_{y,Z}^2(s,\tau) = \dfrac{\underset{W}{\leftrightarrow}^{y,Z}(s,\tau)\, \underset{W}{\leftrightarrow}^{Z,Z}(s,\tau)^{-1}\, \overline{\underset{W}{\leftrightarrow}^{y,Z}(s,\tau)}}{\underset{W}{\leftrightarrow}^{y,y}(s,\tau)}$ (3)
$\qquad R_{x,Z}^2(s,\tau) = \dfrac{\underset{W}{\leftrightarrow}^{x,Z}(s,\tau)\, \underset{W}{\leftrightarrow}^{Z,Z}(s,\tau)^{-1}\, \overline{\underset{W}{\leftrightarrow}^{x,Z}(s,\tau)}}{\underset{W}{\leftrightarrow}^{x,x}(s,\tau)}$ (4)
$\quad \gamma_{y,x}(s,\tau)$ is the complex wavelet coherence between $y$ and $x$, which can be written as
$\qquad \gamma_{y,x}(s,\tau) = \dfrac{\underset{W}{\leftrightarrow}^{y,x}(s,\tau)}{\left(\underset{W}{\leftrightarrow}^{y,y}(s,\tau)\underset{W}{\leftrightarrow}^{x,x}(s,\tau)\right)^{1/2}}$ (5)
where $\underset{(\cdot)}{\leftrightarrow}$ is the smoothing operator, $\overline{(\cdot)}$ is the complex conjugate operator, $(\cdot)^{-1}$
indicates the inverse of the matrix, and
$\qquad \underset{W}{\leftrightarrow}^{y,Z}(s,\tau) = \left[\underset{W}{\leftrightarrow}^{y,Z_1}(s,\tau)\ \underset{W}{\leftrightarrow}^{y,Z_2}(s,\tau)\cdots \underset{W}{\leftrightarrow}^{y,Z_q}(s,\tau)\right]$ (6)
$\qquad \underset{W}{\leftrightarrow}^{x,Z}(s,\tau) = \left[\underset{W}{\leftrightarrow}^{x,Z_1}(s,\tau)\ \underset{W}{\leftrightarrow}^{x,Z_2}(s,\tau)\cdots \underset{W}{\leftrightarrow}^{x,Z_q}(s,\tau)\right]$ (7)
$\qquad \underset{W}{\leftrightarrow}^{Z,Z}(s,\tau) = \begin{bmatrix} \underset{W}{\leftrightarrow}^{Z_1,Z_1}(s,\tau) & \cdots & \underset{W}{\leftrightarrow}^{Z_1,Z_q}(s,\tau) \\ \vdots & \ddots & \vdots \\ \underset{W}{\leftrightarrow}^{Z_q,Z_1}(s,\tau) & \cdots & \underset{W}{\leftrightarrow}^{Z_q,Z_q}(s,\tau) \end{bmatrix}$ (8)
where $\underset{W}{\leftrightarrow}^{A,B}(s,\tau)$ is the smoothed auto-wavelet power spectra (when $A=B$) or cross-
wavelet power spectra (when $A \neq B$) at scale $s$ and location $\tau$, respectively. Please refer to
previous publications for detailed calculation of smoothed auto- and cross-wavelet power
spectra (Grinsted et al., 2004; Hu and Si, 2016).





The squared PWC (hereinafter referred to as PWC) at scale $s$ and location $\tau$ , $\rho_{y,x \cdot Z}^2$,
can be written as
$$\rho_{y,x \cdot Z}^2 = \frac{\left|1 - R_{y,x \cdot Z}^2(s,\tau)\right|^2 R_{y,x}^2(s,\tau)}{\left(1 - R_{y,Z}^2(s,\tau)\right)\left(1 - R_{x,Z}^2(s,\tau)\right)} \qquad (9)$$
where
$$R_{y,x}^2(s,\tau) = \frac{\overleftrightarrow{W}^{y,x}(s,\tau)\overline{\overleftrightarrow{W}^{y,x}(s,\tau)}}{\overleftrightarrow{W}^{y,y}(s,\tau)\overleftrightarrow{W}^{x,x}(s,\tau)} \qquad (10)$$
The phase angle between $y$ and $x$ after excluding effect of $Z$ is
$$\vartheta_{y,x \cdot Z}(s,\tau) = \varphi_{y,x \cdot Z}(s,\tau) + \vartheta_{y,x}(s,\tau) \qquad (11)$$
where
$$\varphi_{y,x \cdot Z}(s,\tau) = \arg\left(1 - R_{y,x \cdot Z}^2(s,\tau)\right) \qquad (12)$$
and $\vartheta_{y,x}(s,\tau)$ is the wavelet phase between $y$ and $x$, which can be expressed as
$$\vartheta_{y,x}(s,\tau) = \tan^{-1}\left(\mathrm{Im}\big(W^{y,x}(s,\tau)\big)/\mathrm{Re}\big(W^{y,x}(s,\tau)\big)\right) \qquad (13)$$
where $\arg$ denote the argument of the complex number, $W^{y,x}(s,\tau)$ is the cross-wavelet
power spectrum between $y$ and $x$ at scale $s$ and location $\tau$; Im and Re denote the
imaginary and real part of $W^{y,x}(s,\tau)$, respectively.
The Monte Carlo method is used to calculate PWC at 95% confidence level. In brief,
calculation of PWC is repeated for a sufficient number of times using data generated by
Monte Carlo simulations based on the first-order autocorrelation coefficient. The 95[th]

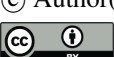



percentile of PWCs of all simulations at each scale represents the PWC at the 95%
confidence level. The average PWC, percent area of significant coherence (PASC) relative
to the whole wavelet scale–location domain, and average value of significant PWC (PWC$_{sig}$)
are also calculated for different scale-location domains. The Matlab codes for calculating
PWC and significance level are provided in the Supplement (Sect. S1–S3).
The new method is compared with the method of Mihanović et al. (2009) in the case of
one excluding variable ($Z = \{Z_1\}$). The Mihanović et al. (2009) method was developed
directly from the traditional partial correlation analysis (Kenney and Keeping, 1939), and
therefore has a similar equation for calculating PWC, which can be expressed as
$$\rho_{y,x \cdot Z1}^2 = \frac{\left| R_{y,x}(s,\tau) - R_{y,Z1}(s,\tau)\, R_{x,Z1}(s,\tau) \right|^2}{\left(1 - R_{y,Z1}^2(s,\tau)\right)\left(1 - R_{x,Z1}^2(s,\tau)\right)} \qquad (14)$$

In the case of one excluding variable, the numerators between Eqs. (9) and (14) differ,
but the denominators remain the same.
**3. Data and analysis**
**3.1 Artificial data for method test**
The PWC is first tested using the cosine-like artificial dataset produced following Yan
and Gao (2007). The cosine-like artificial datasets are suitable for testing the new method
because they mimic many spatial or temporal series in geoscience such as climatic variables,
hydrologic fluxes, seismic signals, El Niño-Southern Oscillation, land surface topography,
ocean waves, and soil moisture. The procedures to test the PWC is largely based on Hu and





Si (2016), where the same dataset has been used to test the MWC method. Please refer to
Hu and Si (2016) for the detailed description of the artificial dataset. In brief, the response
variable ($y$ and $z$ for the stationary and non-stationary case, respectively) is the sum of five
cosine waves ($y_1$ to $y_5$ and $z_1$ to $z_5$ for the stationary and non-stationary case, respectively)
at 256 locations (Hu and Si, 2016). For $y_1$, $y_2$, $y_3$, $y_4$, and $y_5$, they have consistent
dimensionless scales of 4, 8, 16, 32, and 64, respectively, across the series. For $z_1$, $z_2$, $z_3$, $z_4$,
and $z_5$, the dimensionless scales gradually change with location, with the maximum
dimensionless scales of 4, 8, 16, 32, and 64, respectively. The variance of the response
variable $y$ and $z$ is 2.5. All other variables ($y_1$ to $y_5$ or $z_1$ to $z_5$) are orthogonal to each other
with equal variance of 0.5. The predictor and excluding variables (Fig. S1 of Sect. S4 in the
Supplement) are selected from the five cosine waves (e.g., $y_1$ to $y_5$ or $z_1$ to $z_5$) or their
derivatives. The exact variables and procedures to test the new PWC method are explained
later on.
The PWC between response variable $y$ (or $z$) and predictor variable, i.e., $y_2$ (or $z_2$), is first
calculated after excluding the effect of one variable. Four types of excluding variable are
involved (Fig. S1 of Sect. S4 in the Supplement): (a) original series of $y_2$ (or $z_2$) or $y_4$ (or
$z_4$); (b) second half of the original series of $y_2$ (or $z_2$) are replaced by 0 to simulate abrupt
changes (i.e., transient and localized feature) of the spatial series. They are referred to as
$y_2h_0$ (or $z_2h_0$); (c) white noises with zero-mean and standard deviations of 0.3 (weak noise),
1 (moderate noise), and 4 (high noise) are added to $y_2$ (or $z_2$) as suggested by Hu and Si
(2016) to simulate non-perfect cyclic patterns of the excluding variables. They are referred
to as $y_2wn$ (or $z_2wn$), $y_2mn$ (or $z_2mn$), and $y_2sn$ (or $z_2sn$), respectively; and (d) a combination




of type b and type c. They are referred to as $y_2wnh_0$ (or $z_2wnh_0$), $y_2mnh_0$ (or $z_2mnh_0$), and
$y_2snh_0$ (or $z_2snh_0$), respectively. The same data are also analyzed using the Mihanović et al.
(2009) method for comparison.
The PWC between response variable $y$ (or $z$) and predictor variable, i.e., $y_2y_4$ (sum of $y_2$
and $y_4$) for the stationary case or $z_2z_4$ (sum of $z_2$ and $z_4$) for the non-stationary case, is
calculated with two excluding variables, which is a combination of $y_4$ (or $z_4$) and $y_2$ (or $z_2$)
or its noised series ($y_2wn$ or $z_2wn$, $y_2mn$ or $z_2mn$, and $y_2sn$ or $z_2sn$). Note that PWC between
$y$ (or $z$) and other predictor variables (e.g., $y_4$ or $z_4$) after excluding $y_2$ or $z_2$ and their
equivalent derivative variables (i.e., noised variables or variables with 0) are also calculated.
The related results are not shown because they are analogous to those in case of predictor
variable of $y_2$ (or $z_2$).
The merit of the artificial data is that we know the exact scale- and localized bivariate
relationships after the effect of excluding variables is removed. Theoretically, we expect (a)
PWC is 1 at scales corresponding to scale difference of excluding variables from predictor
variable, and 0 at other scales. For example, PWC between $y$ and $y_2y_4$ after excluding the
effect of $y_4$ is expected to be 1 at the scale of 8, which is the difference of $y_4$ (32) from $y_2y_4$
(8 and 32), and 0 at other scales (e.g., 32); (b) PWC remains 1 at the second half of series
where spatial series is replaced by 0, and 0 at the first half of the original series. For example,
PWC between $y$ and $y_2$ after excluding the effect of $y_2h_0$ is expected to be 0 and 1 at the first
and second half of series, respectively, at the scale of 8; and (c) PWC increases as more
noises are included in the excluding variables. For example, PWC between $y$ and $y_2$ after





excluding the effect of noised series of $y_2$ is expected to increase with increasing noises in
an order of $y_2sn>y_2mn>y_2wn$ at the scale of 8.
**3.2 Real data for application**
3.2.1 Free water evaporation
The free water evaporation dataset has been used to test the MWC (Hu and Si, 2016). In
brief, this dataset includes monthly free water evaporation (E), mean temperature (T),
relative humidity (RH), sun hours (SH), and wind speed (WS) between January 1979 and
December 2013 at Changwu site in Shaanxi province provided by the China Meteorological
Administration. During this period, the average daily temperature was 9.4 °C, the average
annual rainfall was 571 mm and annual $ET_p$ was 883 mm. Being located in the transition
between semi-arid and subhumid climates, agricultural production at the Changwu site is
constrained by water availability. The PWC between E and each meteorological variable is
calculated by excluding the effect of each or all of the other meteorological variables.
Results of wavelet power spectrum of E and BWC between every two variables are shown
in Fig. S2 and Fig. S3 (Sect. S5 in the Supplement), respectively.
3.2.2 Soil water content
Soil water datasets were obtained from the hummocky landscape of Canadian Prairies
(Biswas and Si, 2011b; Hu et al., 2017). The sampling site is characterized by a subhumid
continental climate with Dark Brown Chernozem soils. Data were collected from 128
locations with equal intervals (4.5 m) along a 576 m long transect. Soil water contents of





top layer (0–0.2 m) were measured by a portable Tektronix TDR in spring (May 2, 2008)
and summer (August 23, 2008). Other environmental variables measured were clay content,
sand content, soil organic carbon content (SOC), bulk density (BD) of 0–0.2 m, depth to
$CaCO_3$ layer (vertical distance between surface and the layer of first presence of $CaCO_3$),
elevation, slope, aspect (calculated as cos(aspect)), and wetness index. Please refer to
previous studies for detailed information on this dataset (Biswas et al., 2012; Biswas and
Si, 2011a, b).
The PWC between SWC and each environmental variable is calculated by excluding the
effect of another environmental factor. The BWC between SWC and each environmental
factor (Fig. S4 and S5 of Sect. S5 in the Supplement), BWC between environmental factors
(Fig. S6 of Sect. S5 in the Supplement), and MWC between SWC and environmental factors
have been previously analyzed (Biswas and Si, 2011a; Hu et al., 2017).
**4. Results and discussion**
**4.1 PWC with artificial data**
4.1.1   PWC with one excluding variable using the new method
Fig. 1a shows PWC between dependent variable $y$ (or $z$) and predictor variable $y_2$ (or $z_2$)
by excluding one variable. For the stationary case, there is one horizontal band (red color)
representing an in-phase high PWC value at scales around 8 for all locations after
eliminating the effect of $y_4$. Note that the PWC values between $y$ and $y_2$ after excluding the
effect of $y_4$ are not exactly 1 as would be expected at all scale-location domains, because of





the effect of smoothing along scales and locations. However, the PWC values at the center
of the significant band, corresponding to the exact scale (8) of the predictor variable $y_2$, are
very close to 1 (0.996), and the mean $PWC_{sig}$ values are very high (i.e., 0.96). The result is
similar to the BWC between $y$ and $y_2$. This is understandable because $y_4$ is orthogonal to $y_2$,
and excluding the effect of $y_4$ does not affect the relationship between $y$ and $y_2$ at all.

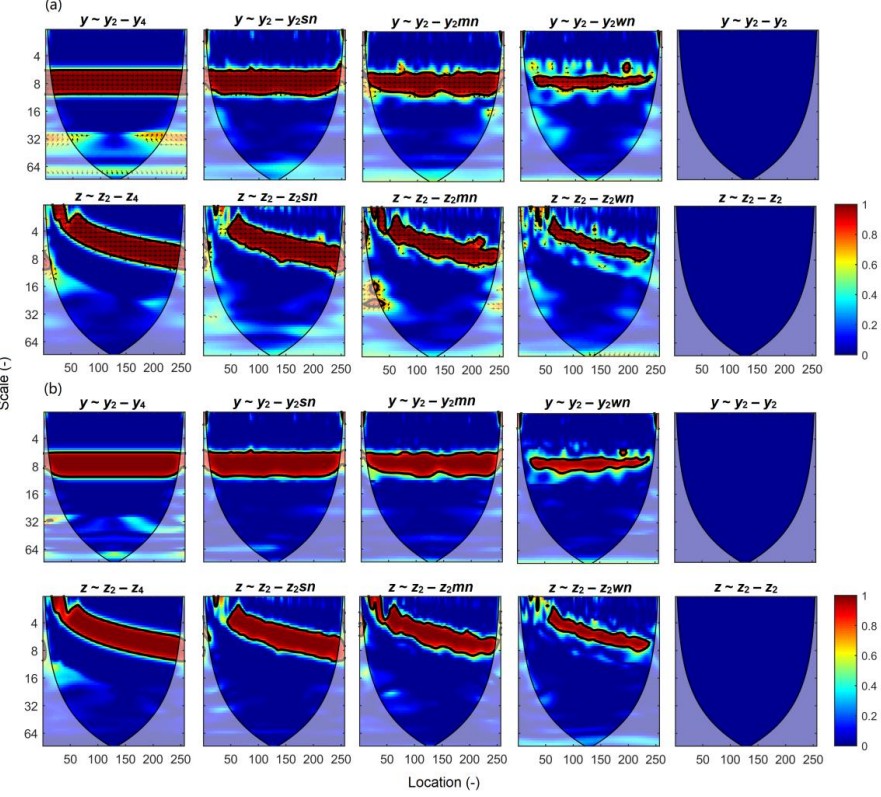


**Figure 1.**
Partial wavelet coherency (PWC) between response variable $y$ (or $z$) and predictor variable
$y_2$ (or $z_2$) after excluding the effect of variables $y_4$ (or $z_4$), $y_2sn$ (or $z_2sn$), $y_2mn$ (or $z_2mn$),
$y_2wn$ (or $z_2wn$), and $y_2$ (or $z_2$) for the stationary (or non-stationary) case using the new





method (a) and Mihanović et al. (2009) method (b). Arrows represent the phase angles of the cross-wavelet power spectra between two variables after eliminating the effect of excluding variables. Arrows pointing to the right (left) indicate positive (negative) correlations. Thin and thick solid lines show the cones of influence and the 95% confidence levels, respectively. All variables are explained in Section 3.1 and are shown in Fig. S1 of Sect. S4 in the Supplement.

Similar results were obtained by excluding either $y_4$ or the strongly noised series of $y_2$ ($y_2sn$). Compared with the case of excluding variable of $y_4$, excluding the effect of $y_2sn$ results in slightly narrower band of significant PWC and slightly reduced mean $PWC_{sig}$ (0.94 versus 0.96). When less noise is included in the excluding variables (i.e., $y_2mn$ and $y_2wn$), the significant PWC band becomes narrower. The PASC values are 86%, 77%, and 32% for excluding $y_2sn$, $y_2mn$ and $y_2wn$, respectively, at scales of 6–10. Moreover, the mean $PWC_{sig}$ decreases from 0.94 ($y_2sn$) to 0.93 ($y_2mn$) and 0.89 ($y_2wn$) when progressively more noise is added (Fig. 1a). If we exclude the predictor variable $y_2$ itself, there are, as we expect, no correlations between $y$ and $y_2$ (Fig. 1a). For the non-stationary case, similar results are obtained (Fig. 1a). The only difference is that the scales with significant PWC values change with location, as is found for MWC (Hu and Si, 2016).



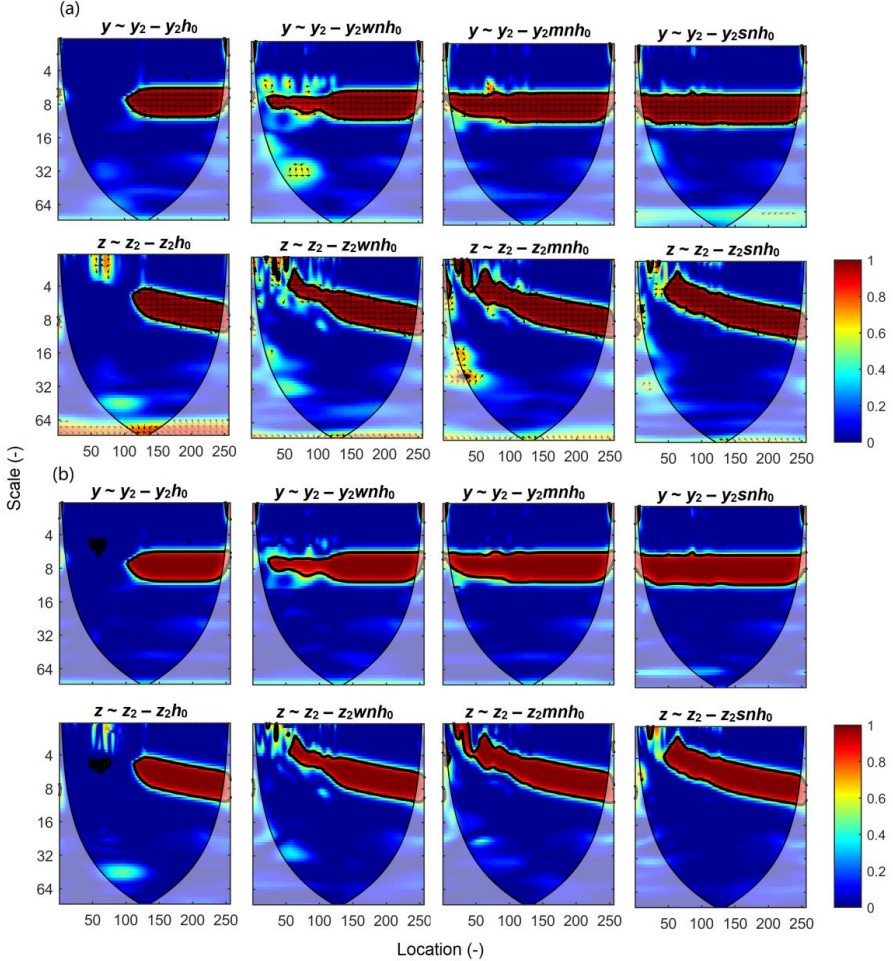

**Figure 2.**

Partial wavelet coherency (PWC) between response variable $y$ (or $z$) and predictor variable $y_2$ (or $z_2$) after excluding effect of variables $y_2h_0$ (or $z_2h_0$), $y_2wnh_0$ (or $z_2wnh_0$), $y_2mnh_0$ (or $z_2mnh_0$), and $y_2snh_0$ (or $z_2snh_0$), for the stationary (or non-stationary) case using the new method (a) and Mihanović et al. (2009) method (b). All variables are explained in Section 3.1 and are shown in Fig. S1 of Sect. S4 in the Supplement.

When second half of the excluding variable series is replaced by 0, the PWC values in that half are close to 1, while those in the first half of data series are 0 at scales





corresponding to the predictor variable (Fig. 2a). For the stationary case, after excluding
the effect of $y_2h_0$, the PWC values are close to 1 (0.98) and 0 in the second and first half of
the data series, respectively, at the dimensionless scale of 8 (Fig. 2a). Similar results are
observed for the non-stationary case (Fig. 2a). This is anticipated because removing series
of 0s from a portion of the predictor variable series does not affect their correlations at these
locations. If different magnitudes of noises are added to the first half of the excluding
variables ($y_2$ or $z_2$), the significant PWC band in the first half becomes wider as the
magnitude of noises increases, while the significant PWC band in the second half remains
almost unchanged. Take the stationary case for example, the PASC values at scales of 6–10
are 40% ($y_2wnh_0$), 74% ($y_2mnh_0$), and 86% ($y_2snh_0$) in the first half, respectively, while those
values vary from 86% to 90% in the second half. Meanwhile, the mean $PWC_{sig}$ in the first
half at scales of 6–10 increases from 0.91 to 0.94 in both the stationary and non-stationary
cases as more noises are added to the excluding variable $y_2$ or $z_2$. This indicates that the new
PWC method can also capture the abrupt changes in the data series, and has the ability to
deal with localized relationships.
4.1.2  PWC with two excluding variables using the new method
When both $y_2$ and $y_4$ (or $z_2$ and $z_4$) are considered in the predictor variables, there are two
bands of wavelet coherence of 1 between $y$ (or $z$) and $y_2y_4$ (or $z_2z_4$), which correspond to the
scales of two predictor variables (Hu and Si, 2016). However, after the effect of $y_4$ (or $z_4$)
is removed, only one band with PWC of around 1 occurs at the scale of the predictor
variable $y_2$ (or $z_2$) (Fig. 3), which is identical to the PWC between $y$ (or $z$) and $y_2$ (or $z_2$) after





excluding the effect of variable $y_4$ (or $z_4$) (Fig. 1). After both predictor variables $y_2$ and $y_4$
(or $z_2$ and $z_4$) are excluded, the PWC between $y$ (or $z$) and $y_2y_4$ (or $z_2z_4$) is 0 at all scale-
location domains as we expect. When one of the excluding variables $y_2$ (or $z_2$) is added with
noises, the relationship between response variable $y$ (or $z$) and predictor variable $y_2y_4$ (or
$z_2z_4$) becomes significant at scales of the excluding variable $y_2$ (or $z_2$). Similar to the case of
one excluding variable (Fig. 1), less noise in the excluding variable of $y_2$ (or $z_2$) results in
narrower significant PWC band, and reduced mean $PWC_{sig}$ values (from 0.96 ($y_2sn$) to 0.90
($y_2wn$) in the stationary case and from 0.95 ($z_2sn$) to 0.92 ($z_2wn$) in the non-stationary case)
(Fig. 3).

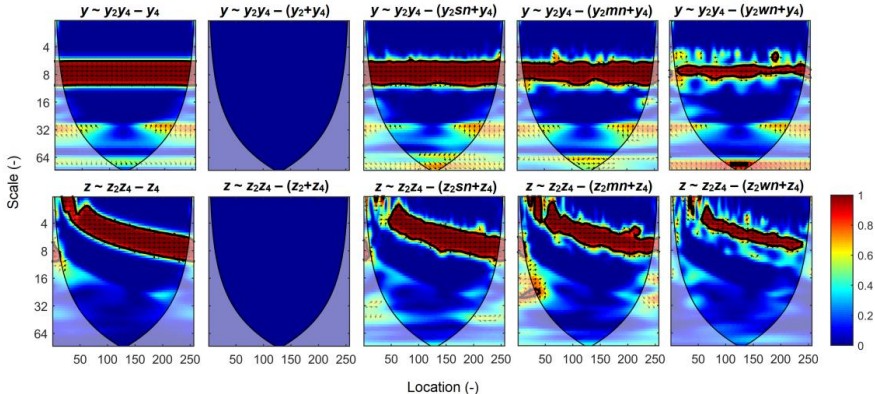

**Figure 3.**
Partial wavelet coherency (PWC) between response variable $y$ (or $z$) and predictor variable
$y_2y_4$ (or $z_2z_4$) after excluding the effect of variables $y_4$ (or $z_4$), $y_2+y_4$ (or $z_2+z_4$), $y_2$sn+$y_4$ (or
$z_2sn+z_4$), $y_2mn+y_4$ (or $z_2mn+z_4$), and $y_2wn+y_4$ (or $z_2wn+z_4$) for the stationary (or non-
stationary) case. All variables are explained in Section 3.1 and are shown in Fig. S1 of Sect.
S4 in the Supplement.





### 4.1.3  Comparison of the new method with the Mihanović et al. (2009) method


In the case of one excluding variable, the corresponding PWC values calculated with the
Mihanović et al. (2009) method are shown in Figs 1b and 2b. Except for the phase
information, the two methods generally produce comparable coherence despite the differing
numerators in their corresponding equations (Eq. 9 and 14). However, we notice that the
new PWC method produces consistently slightly higher coherence than the Mihanović et
al. (2009) method. For example, their mean PWCs between $y$ and $y_2$ at the scale of 8 after
excluding the effect of $y_4$ are 1.00 and 0.97, respectively. This may indicate that the new
method slightly outperforms the Mihanović et al. (2009) method because we expect that the
coherence between $y$ and $y_2$ at the scale (8) of $y_2$ is exactly 1.
Note that some unexpected high PWC can be produced in some domains by the new
method. For example, at a scale of 32, PWC values between $y$ and $y_2$ after excluding $y_4$ are
not significant, but relatively high, partly because of small octaves (default of 1/12) per
scale. This spurious unexpected high PWC is caused by low values in both the numerator
(partly associated with the low coherence between response $y$ and predictor variables $y_2$ at
scale of 32) and denominator (partly associated with the high coherence between response
$y$ and excluding variable $y_4$ at a scale of 32) in Eq. (9). The same problem also exists in the
Mihanović et al. (2009) method (Fig. 1b and 2b). Particularly, the Mihanović et al. (2009)
method produces some positive infinite coherence (small black zones) between $y$ (or $z$) and
$y_2$ (or $z_2$) after eliminating the effect of $y_2h_0$ (or $z_2h_0$) (Fig. 2b) because of extremely low
values in the both numerator and denominator term in Eq. (14). However, it seems that the





domain with overestimation by the new method is very limited and it is located mainly
outside of the cones of influence. Anyway, the unexpected results can be easily ruled out
with knowledge of BWC between response and predictor variables.
Compared with the Mihanović et al. (2009) method, our new PWC method can be used
to deal with situations with more than one excluding variable, which is a knowledge gap.
Moreover, inclusion of phase information in the new PWC is another advantage of this
method.
**4.2 PWC with real data**
4.2.1   Free water evaporation
The PWC analysis indicates that the correlations between E and T after excluding the
effect of each of other three variables (RH, SH, and WS) were almost the same as those
indicated by the BWC (Fig. 4 and Fig. S3 of Sect. S5 in the Supplement). For example,
after excluding the effect of RH, E and T were positively correlated at the medium scales
(8–32 months). The PASC was 61% and mean $PWC_{sig}$ value was 0.94, which was identical
to the case of BWC between E and T. The significant correlations at scales around 64
months between E and T from 1979 to 1992 were absent after eliminating the influence of
RH. This implies that the influence of mean temperature on $E$ at these scales and years may
be associated with the negative influence of RH on both E and T (Fig. S3 of Sect. S5 in the
Supplement).

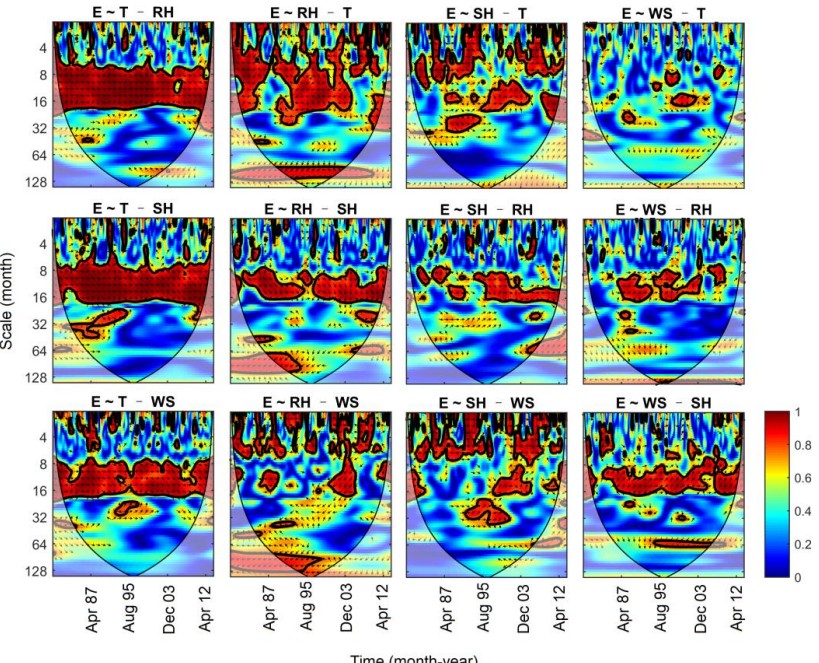

**Figure 4.**

Partial wavelet coherency (PWC) between evaporation (E) and each meteorological factor
(T, mean temperature; RH, relative humidity; SH, sun hours; WS, wind speed) after
excluding the effect of each of other three meteorological factors.

The PWC between E and RH depended on the excluding variable and scale (Fig. 4). The
mean PWC and PASC between E and RH after excluding T were 0.60 and 34%, respectively,
which are comparable to the mean BWC (0.62) and PASC (40%) between E and RH. The
corresponding values after excluding SH and WS were 0.50 and 0.53 (PWC), 22% and 21%
(PASC), respectively. In addition, compared with the BWC between E and RH, correlations
between E and RH were almost absent at small scales (<8 months) and medium scales (8–
32 months) after eliminating the influence of SH and WS, respectively. Therefore,





excluding variable of T had less influence on the coherence between E and RH compared
with excluding variables of SH and WS. This is mainly because relative humidity and
temperature are correlated with E at different scales (Fig. S3 of Sect. S5 in the Supplement),
i.e., mean temperature affected E mainly at medium scales, while RH affected E across all
scales. However, the domain where SH and WS were correlated with E was subset of that
where RH and E were correlated (Fig. 4).

The relationships between E and sun hours after excluding other three factors were less

consistent. The areas with significant corrections were scattered over the whole frequency-
time domain but differed with excluding factors. The PASC varied from 12% (excluding
RH) to 20% (excluding T and WS), which is much lower than the PASC (28%) in the case
of BWC. The significant relationships between E and WS were only limited to very small
areas except for the case of SH being excluded, where E and wind speed were positively
correlated at scales of 8–16 months most of the time.

In general, the PASC decreased after excluding the effects of more factors (data not

shown). The correlations between E and each variable after eliminating the effects of all
other variables are shown in Fig. 5. The correlations between E and T were still significant
at the medium scales (8–32 months), where PASC value was 52% with mean $\text{PWC}_{\text{sig}}$ of
0.92. The E was still correlated with RH at large scales (>32 months), where PASC value
was 35% with mean $\text{PWC}_{\text{sig}}$ of 0.96. Interestingly, the domain with significant correlation
between E and SH and WS was very limited. This indicates that the influences of SH and
WS on E have already been covered by RH and T. This is in agreement with the MWC





results that RH and T were the best to explain E variations at all scales. Although the RH
had the greatest mean wavelet coherence and PASC at the entire scale-location domains,
the PWC analysis seems to support that mean temperature was the most dominating factor
for free water evaporation at the 1-year cycle (8–16 months), which is the dominant scale
of E variation (Fig. S2 of Sect. S5 in the Supplement). This further verifies the suitability
of the Hargreaves model (only air temperature and incident solar radiation required)
(Hargreaves, 1989) for estimating potential evapotranspiration on the Chinese Loess
Plateau (Li, 2012).

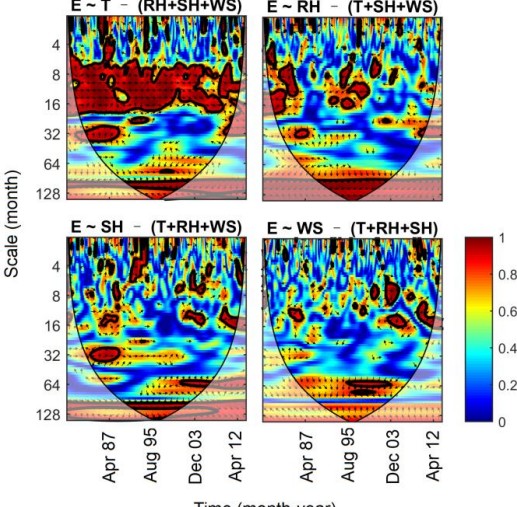


**Figure 5.**
Partial wavelet coherency (PWC) between evaporation (E) and each meteorological factor
(T, mean temperature; RH, relative humidity; SH, sun hours; WS, wind speed) after
excluding the effects of all other three factors.





4.2.2   SWC

In spring, SWC at 0–0.2 m was significantly correlated with elevation, wetness index,

depth to $CaCO_3$ layer, and SOC at large scales (72–144 m); it was significantly correlated
with sand content, SOC, depth to $CaCO_3$ at medium scales (36–72 m) and bulk density at
scales of 36–144 m in the first half of the transect (Fig. S4 of Sect. S5 in the Supplement).
The PWC shows that SWC was not correlated with elevation after eliminating the effect of
SOC or depth to $CaCO_3$ (Fig. 6). By contrast, after the removal of the elevation's effect,
SWC was significantly correlated with SOC at scales of 36–144 m in the first half of the
transect and significantly correlated with depth to $CaCO_3$ layer at large scales (>100 m)
across the transect (Fig. 6). There were little correlations between SWC and wetness index
after eliminating the effect of elevation (Fig. 6). Therefore, the influences of elevation and
wetness index on SWC in spring might have been taken into account by SOC and depth to
$CaCO_3$ layer. Although elevation and wetness index are important drivers of snowmelt run-
off in spring (Hu et al., 2017), they did not contribute any more to explaining SWC
variations than SOC or depth to $CaCO_3$ layer did. The same holds for bulk density and sand
content whose influences on SWC were also limited after eliminating the effect of SOC
(Fig. 6). This was because SOC was negatively correlated with sand content at medium
scales (36–72 m) and bulk density at scales of 36–144 m in the first half of the transect (Fig.
S5 of Sect. S5 in the Supplement). Interestingly, the significant correlations between SWC
and SOC or depth to $CaCO_3$ layer still existed no matter what the excluding factors were.
For example, SWC was significantly correlated with depth to $CaCO_3$ layer at scales >130
m after the effect of SOC was removed; SWC was significantly correlated with SOC at



large scales (>130 m) across the transect and at scales of 36–90 m at locations from 45 to
200 m after eliminating the effect of depth to CaCO$_3$ layer (Fig. 6). This further validates
that the combination of depth to CaCO$_3$ layer and SOC were the best to explain SWC
variations in spring (Hu et al., 2017).

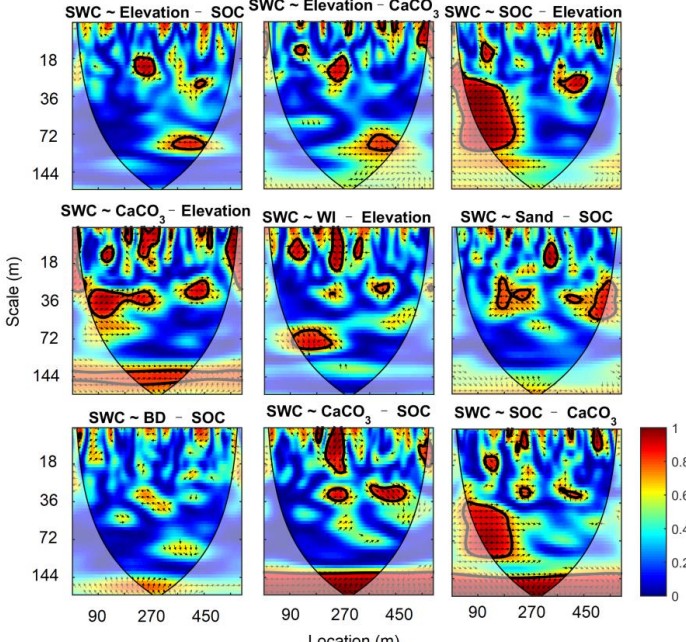


**Figure 6.**


Partial wavelet coherency (PWC) between soil water content (SWC) in spring and one
environmental factor after excluding the effect of another environmental factor. SOC, soil
organic carbon; CaCO$_3$, depth to the CaCO$_3$ layer; WI, wetness index; BD, bulk density.

In summer, SWC of 0–0.2 m tended to be significantly affected by aspect, slope,

elevation, wetness index, clay, and sand at large scales (>90 m or 72–144 m) and by SOC,
bulk density, and slope at medium scales (36–72 m) at locations from 45 to 450 m over the





transect (Fig. S5 of Sect. S5 in the Supplement). The PWC analysis indicates that elevation,
wetness index, sand (not shown), clay, and BD had little influences on SWC after
eliminating the effect of slope in summer (Fig. 7). This is largely because slope was
significantly correlated to BD at medium scales and to elevation, wetness index, sand, and
clay at large scales (Fig. S6 of Sect. S5 in the Supplement). However, the influence of slope
on SWC was also limited after eliminating the effect of SOC (Fig. 7). By contrast, the effect
of SOC on SWC at the medium scales still existed at some locations after eliminating the
effects of slope and aspect (Fig. 7). This highlights the dominant role of SOC as a surrogate
of vegetation in driving evapo-transpiration loss at the slope (medium) scales (Hu et al.,
2017). As we expect, the effect of SOC on SWC at the medium scales disappeared after
eliminating the effect of BD because of the strong correlations between SOC and BD (Fig.
S5 of Sect. S5 in the Supplement). However, the effect of SOC on SWC was amplified at
large scales (>72 m) after excluding the effect of BD as also found in the artificial datasets
(Fig. 7). Interestingly, the significant correlation between SWC and aspect at large scales
(>90 m) persisted regardless the excluding variables (as an example, only PWC for
excluding variable of SOC is shown in Fig. 7). This highlights the dominant role of aspect
in driving soil water distribution at large scales in summer. Overall, the PWC analysis
further confirms that a combination of aspect and SOC was the best to explain SWC
variations in summer (Hu et al., 2017).



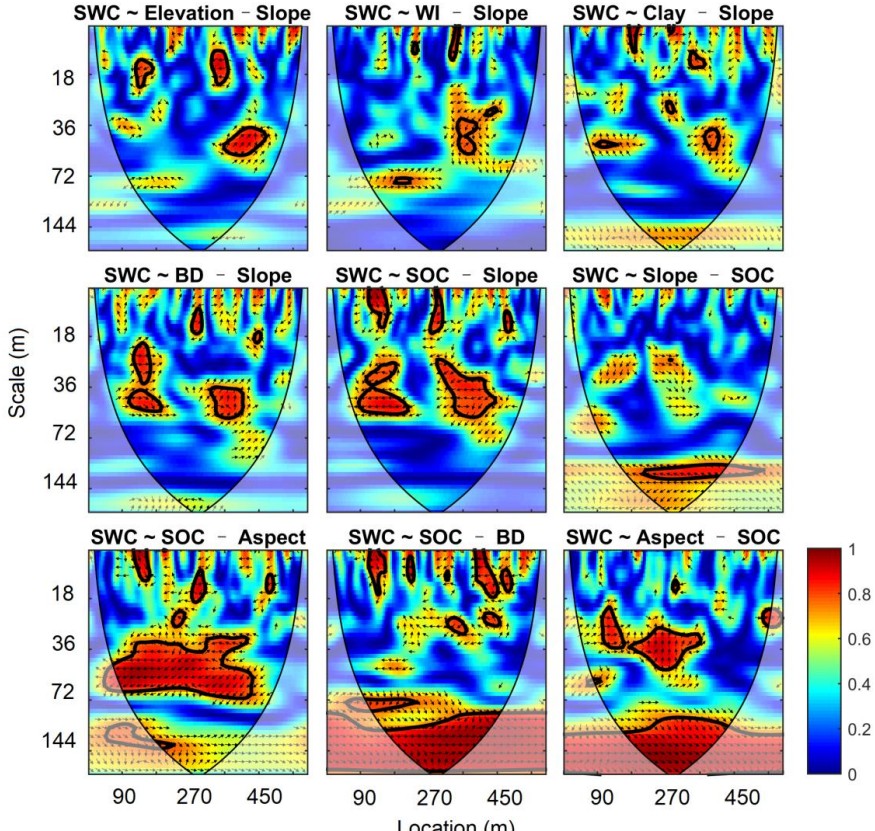

**Figure 7.**

Partial wavelet coherency (PWC) between soil water content (SWC) in summer and one

environmental factor by excluding another environmental factor. SOC, soil organic carbon;

Aspect, Cos(Aspect); WI, wetness index; BD, bulk density.

## 5. Conclusions

Partial wavelet coherency (PWC) is developed in this study to investigate scale-and

location-specific bivariate relationships after excluding the effect of one or more variables

in geosciences. This method was developed on the basis of partial coherence in the



multivariate spectral case (Koopmans, 1995), and is an extension of previous work on PWC and WMC (Hu and Si, 2016; Mihanović et al., 2009). Compared with the previous PWC method (Mihanović et al., 2009), this new method produces slightly more accurate coherence. In addition, the new PWC method has the advantage of dealing with more than one excluding variable and providing the phase information associated with the PWC.

The new PWC method has been successfully tested with the artificial datasets. As we expect, regardless of the stationary and non-stationary case, there are no or reduced correlations between response and predictor variables in scale-location domains where the excluding variables are significantly correlated with the response variable. The new method also has the ability to deal with localized relationships. The new method was applied to two previously published datasets. The application has shown that the coherency between response and predictor variables was less affected by excluding other variables if the predictor variable had dominating roles in explaining the variations in the response variable. This application further confirmed the best combinations for explaining temporal variations in free water evaporation at the Changwu site in China and spatial variations in soil water content in the hummocky landscape in Saskatchewan, Canada.

Like the Mihanović et al. (2009) method (a previous PWC method), the new method has the risk to produce spurious correlations after excluding the effect from other variables. But this spurious high coherence can be easily identified with knowledge of BWC. So, caution should be taken to interpret those results. Similar to BWC and MWC, the new PWC also suffers from the multiple-testing problem (Schaefli et al., 2007; Schulte et al., 2015).



Therefore, the new method can benefit from a better statistical significance testing method.
Our artificial datasets and two real-world datasets have verified that our PWC method
provides an effective tool to untangle the bivariate relationships at multiple scale-location
domains after eliminating the effects of other variables. The new method provides a much
needed data-driven tool for unraveling underlining mechanisms in a spatial or temporal
series. Thus, combining with wavelet transform, BWC, and MWC, the new PWC method
can be used to detect various processes in geosciences, such as stream flow, droughts,
greenhouse gas emissions (e.g., $N_2O$, $CO_2$, and $CH_4$), atmospheric circulation, and oceanic
processes (e.g., EI Niño-Southern Oscillation).
**Acknowledgements**
The Matlab codes for calculating partial wavelet coherency are available in the Supplement
(Sect. S1–S3). The codes are developed based on those provided by Aslak Grinsted
(http://www.glaciology.net/wavelet-coherence)  and  Wei  Hu  and  Bing  Si
(https://www.hydrol-earth-syst-sci.net/20/3183/2016/hess-20-3183-2016-supplement.pdf).
The preparation of this manuscript was partly supported by The New Zealand Institute for
Plant and Food Research Limited under the *Sustainable Agro-ecosystems* programme.

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
