# Peer review of "Technical Note: Partial wavelet coherency for improved understanding of scale-specific and localized bivariate relationships in geosciences"

_Hydrology and Earth System Sciences, 2020_

## Referee Comment (RC1) · Anonymous Referee #1 · 10 Sep 2020

In this paper, the authors mainly developed a partial wavelet coherency method, for identifying the relationship between variables. It is an important issue but also a difficult problem for geo-data analysis, and the method developed would be helpful for the data analysis in geosciences. The following comments are suggested to be considered for further improving the paper: (1) In lines 108-110: the "sufficient number" should be clarified, as it has a big influence on the uncertainty estimation, that is, what number is sufficient? Furthermore, the reason of using first-order autocorrelation coefficient for MC simulation should be explained and discussed. (2) Lines 121-122, some theoretical lines can be provided to show the difference between Eq. (9) and Eq. (14). (3) Regarding the structure, is it more suitable to reorganize the Section 3 and 4, that is,

the artificial data and their results are analyzed and discussed in Section 3, while those of real data are analyzed and discussed in Section 4?

---

## Referee Comment (RC2) · Anonymous Referee #2 · 11 Sep 2020

**Review for manuscript "Technical note: Partial wavelet coherency for improved understanding of scale-specific and localized bivariate relationships in geosciences"**

**Authors:** Wei Hu and Bing Si

**Journal:** Hydrology and Earth System Sciences Discussions

**Summary**

In this technical note, the authors propose a method for identifying relationships between two variables for the case where the two variables are correlated to other variables themselves. They apply their 'updated partial wavelet coherency' (PWC) method to a synthetic dataset and two real-world applications and show that this updated PWC model shows similar performance as existing PWC models. They conclude that their model outperforms existing models because it provides phase information and allows for excluding several correlated variables from the PWC.

**General remarks**

I think that the study addresses a question of interest to the hydrological community, i.e. 'how can we identify the most important driving variables of a certain phenomena at different time scales'. The technical note is generally well structured. However, I think that it lacks a didactical and detailed introduction to the topic, problem, and wavelet analysis. The introduction would significantly benefit from providing examples of when the identification of bivariate relationships are important (i.e. providing a motivation for the study), an in-depth introduction to wavelet analysis (for the readers who are not yet too familiar with the topic), and an introduction to the terminology used. Extending the introduction will increase the length of the note and I suggest removing the practical example number 2 instead. I think it does not provide additional insights regarding the performance of the method proposed compared to the statements that were already made based on the synthetic data and the first practical example. Since the new method does not seem to clearly outperform existing methods, I would better explain why adding phase information and excluding several confounding variables is beneficial for the analysis. I would also add a more detailed discussion of model weaknesses, especially the implications of detecting spurious correlations. In addition, the note would profit from careful language editing.

**Major points**

1. **Abstract:** The abstract is not very accessible to non-wavelet-specialists. I would provide a short example for when such an analysis would be necessary/beneficial and shortly summarize what wavelet coherence analysis is all about. Please also shortly explain why PWC has been introduced in the first place (l. 12). I would also mention the datasets used for model evaluations (l. 14). I think the statement 'producing more accurate results' (l. 18) needs justification, otherwise it is not very credible. I would exclude lines 21-24 because this is a technical note and specific results regarding the example applications going beyond model performance are in my opinion not of interest here.

2. **Introduction:** The introduction should in my opinion provide a motivation for the use of PWC methods, also for non-specialists on the topic e.g. by providing examples of important bivariate relationships in the geosciences and why we may be interested in them. In addition, an introduction to wavelet analysis in general and wavelet coherence analysis in particular should be provided. The reader should also be made familiar with the terminology used, e.g. what kind of scales are you talking about and what is an 'excluding variable'. A clear motivation for why excluding variables and including phases matters is required to underline the benefits of the methods later on in the results and conclusions sections (l. 57-58). Currently, the introduction does not very well prepare readers for what they are going to read in the methods and results sections.

3. **Theory:** I think that you should start even simpler here and provide a short introduction to wavelet analysis (difference between discrete and complex, terminology) and wavelet coherence analysis. In addition, it is unclear to me what exactly the difference between classical PWC and your proposed method is (l. 74-76). Currently, it is not entirely clear to me how the Monte Carlo experiment was performed (l. 108-110). Could you please slightly expand this section?

4. **Data and analysis:** I would recommend removing the 'soil water content' example (section 4.2.2) because as I can see it does not show anything that has not yet been shown by the 'free evaporation example' in terms of the validity of the model. I would rather invest the space in extending the introduction as outlined in more details above. In the figure captions, I would add a reference to the dataset used to generate it. In addition, I am not sure what you would like to show with the cases where the variable of interest is excluded. I would therefore exclude the results referring to this exercise (e.g. Figure 1 last row and see l. 236-237). I also think that the figures would profit a lot from using labels for subfigures, which would facilitate orientation. To me, the difference between the Mihanovic et al. (2009) model and the proposed model are not evident by looking at the Figures presented (a difference of 0.03 does not seem to be a lot, l.293). Therefore, I think the actual advantages of using this new method should better be worked out and explained before a statement such as 'the new method outperforms the Mihanovic et al. method' (l. 293-294) is made. Please also explain why the inclusion of 'phase information' is an advantage of the new method (l. 312-313).

5. **A proper discussion section is missing:** I would add an in-depth discussion of the weaknesses and benefits of the approach and put the new method into perspective by comparing it to existing methods.

6. **Conclusions:** Given the evidence provided in the results section, statements such as 'the new method produces slightly more accurate coherence' do not seem to be justified. As mentioned earlier the benefits of including phase information and excluding several variables need to be better explained. Some of the material presented in this section could be moved to the new discussion section.

7. **Code availability:** I would provide the Matlab code via a data/file repository such as HydroShare or Zenodo instead of the supplement (l.27). This would be very helpful for the community and potential users.

**Minor points**

- L. 31: please explain what you mean by 'time and space localization'.

- L.34: 'among these methods'
- Transition from l. 42 to l. 43: very sharp transition from bivariate relationships to prediction. I would try to establish a clear link between the two things.
- L. 48: what do you mean by 'this issue'?
- L. 50: what kind of scales? Temporal or spatial?
- L. 53-54: would combine greenhouse gas emissions and climate in one category.
- L. 61: information 'which will allow to....'
- L. 61: what do you mean by 'analogy' in this context. I think that rephrasing may be required.
- L. 62: Be specific with what you mean by 'it': 'the proposed method'.
- L. 76: Please explain to the reader what you mean by 'scale' and 'location'.
- L. 99: same for 'phase angle'.
- L. 184-185: can in my opinion be removed.
- L. 191: what does data refer to? Soil water content?
- L. 214: 'significance band'.
- L. 215-216: is this statement underlined by any analysis performed?
- L. 247: what is the purpose of replacing half of the time series by 0?
- L. 261-263: Which feature in the plots actually indicates these 'abrupt changes'?
- L. 266: I can only see one wavelet band of high significance in Figure 3. Where is the second one you mention here?
- L. 298: introduce term 'octave'.
- L. 363-366: would move this sentence to discussion section.

---

## Referee Comment (RC3) · Anonymous Referee #3 · 28 Sep 2020

Partial wavelet coherency for improved understanding of scale-specific and localized bivariate relationships in geosciences Wei Hu and Bing Si

Comments In this paper, the authors presented an improved variant of PWC for identifying the relationship between variables. This should be reflected in the title (like Improved PWC etc to be included in the title) to convey novel contribution. Also at present it is misleading like the authors proposes PWC concept

Overall the paper is well written. I recommend for minor revision

• Line 18– and producing more accurate results.- pl give quantitative statements
• Line 31- provide the developments in chronological order – should be checked at

all places  c What is the real advantage in bringing the phase information in practical cases ? this should be mentioned in the introduction section  c Line 109 .. sufficient number of times using . . .pl make it clear  c Line 214- significance band  c Conclusion: Avoid the statements like – 'this new method produces slightly more accurate coherence'  c Line 450-455 should be explained better ; how can you overcome such problems ? I think better to provide a discussion section before conclusion where such references and unfamiliar terms can be explained in a better way. Then conclusion section should be presented as more specific

———————————————

---

## Author Comment (AC1) · 7 Oct 2020

**Response to Anonymous Referee #1**

Comments from Referee #1

*In this paper, the authors mainly developed a partial wavelet coherency method, for identifying the relationship between variables. It is an important issue but also a difficult problem for geo-data analysis, and the method developed would be helpful for the data analysis in geosciences. The following comments are suggested to be considered for further improving the paper:*

***Comment #1:***
*(1) In lines 108-110: the "sufficient number" should be clarified, as it has a big influence on the uncertainty estimation, that is, what number is sufficient? Furthermore, the reason of using first-order autocorrelation coefficient for MC simulation should be explained and discussed.*

***Response #1:***
Many thanks for your review and positive general comment.
To address the "sufficient number" issue, different combinations of r1 (first-order autocorrelation coefficient) values (i.e., 0.0, 0.5, and 0.9) were used to generate 10 to 10 000 AR(1) series with three, four and five variables. Our results indicate that the noise combination has little impact on the PWC values at the 95% confidence level as also found by Grinsted et al. (2004) for the BWC case (data not shown). The relative difference of PWC at the 95% confidence level compared to that calculated from 10 000 AR(1) series decreases with increase in number of AR(1) series. When the number of AR(1) is above 300, very low maximum relative difference (e.g., <2%) is observed (Fig. RC1 which will be put in the Supplement as Fig. S1 of Sect. S3). Therefore, a replication of 300 seems to be efficient for significance test. If calculation time is not a barrier, however, greater replication number such as ≥1000 is recommended. This will be added into the revision.

[Figure]

**Figure RC1**. Relationship between maximum relative difference (%) of PWC compared to that calculated from 10 000 AR(1) series (surrogate dataset) versus the number of AR(1) series during the significance test using the Monte Carlo test. Number of scales per octave is 12.

The first-order autocorrelation coefficients (r1) in brackets refer to those for the response variable (first), predictor variable (second), and excluding variables (third and onwards).

"The first-order autoregressive model (AR(1)) is chosen because it can be used to simulate most geoscience data very well (Grinsted et al., 2004; Si and Farrell, 2004; Wendroth et al., 1992)"

***Comment #2:***

*(2) Lines 121-122, some theoretical lines can be provided to show the difference between Eq. (9) and Eq. (14).*

***Response #2:***
The difference between Eq. (9) and Eq. (14) will be explained by derivation of PWC in the case of one excluding variable from Eq. (1).

So, when only one variable (e.g., $Z1$) is excluded, Eq.(9) ($\rho_{y,x \cdot Z}^2 = \frac{\left|1 - R_{y,x \cdot Z}^2(s,\tau)\right|^2 R_{y,x}^2(s,\tau)}{\left(1 - R_{y,Z}^2(s,\tau)\right)\left(1 - R_{x,Z}^2(s,\tau)\right)}$ )

can be written as

$$\rho_{y,x \cdot Z1}^2 = \frac{\left|1 - R_{y,x \cdot Z1}^2(s,\tau)\right|^2 R_{y,x}^2(s,\tau)}{\left(1 - R_{y,Z1}^2(s,\tau)\right)\left(1 - R_{x,Z1}^2(s,\tau)\right)} \tag{RC1}$$

Based on equations (2) in our paper,

$$\rho_{y,x \cdot Z1}^2 = \frac{\left|1 - \frac{\overset{\leftrightarrow}{W}^{y,Z1}(s,\tau)\,\overset{\leftrightarrow}{W}^{Z1,Z1}(s,\tau)^{-1}\,\overline{\overset{\leftrightarrow}{W}^{x,Z1}(s,\tau)}}{\overset{\leftrightarrow}{W}^{y,x}(s,\tau)}\right|^2 \frac{\left|\overset{\leftrightarrow}{W}^{y,x}(s,\tau)\right|^2}{\overset{\leftrightarrow}{W}^{y,y}(s,\tau)\,\overset{\leftrightarrow}{W}^{x,x}(s,\tau)}}{\left(1 - R_{y,Z1}^2(s,\tau)\right)\left(1 - R_{x,Z1}^2(s,\tau)\right)}$$

$$= \frac{\left|\overset{\leftrightarrow}{W}^{y,x}(s,\tau) - \frac{\overset{\rightarrow}{W}^{y,Z1}(s,\tau)\overline{\overset{\rightarrow}{W}^{x,Z1}(s,\tau)}}{\overset{\leftrightarrow}{W}^{Z1,Z1}(s,\tau)}\right|^2}{\overset{\leftrightarrow}{W}^{y,y}(s,\tau)\overset{\rightarrow}{W}^{x,x}(s,\tau)\left(1 - R_{y,Z1}^2(s,\tau)\right)\left(1 - R_{x,Z1}^2(s,\tau)\right)}$$

$$= \frac{\frac{1}{\sqrt{\left(\overset{\leftrightarrow}{W}^{y,y}(s,\tau)\overset{\rightarrow}{W}^{x,x}(s,\tau)\right)^2}}\left|\overset{\leftrightarrow}{W}^{y,x}(s,\tau) - \frac{\overset{\leftrightarrow}{W}^{y,Z1}(s,\tau)\overline{\overset{\leftrightarrow}{W}^{x,Z1}(s,\tau)}}{\left(\sqrt{\overset{\rightarrow}{W}^{Z1,Z1}(s,\tau)}\right)^2}\right|^2}{\left(1 - R_{y,Z1}^2(s,\tau)\right)\left(1 - R_{x,Z1}^2(s,\tau)\right)}$$

$$= \frac{\left| \dfrac{\underset{W}{\leftrightarrow}^{y,x}(s,\tau)}{\sqrt{\underset{W}{\leftrightarrow}^{y,y}(s,\tau)}\sqrt{\underset{W}{\leftrightarrow}^{x,x}(s,\tau)}} - \dfrac{\underset{W}{\leftrightarrow}^{y,Z1}(s,\tau)\ \overline{\underset{W}{\leftrightarrow}^{x,Z1}(s,\tau)}}{\sqrt{\underset{W}{\leftrightarrow}^{y,y}(s,\tau)}\sqrt{\underset{W}{\leftrightarrow}^{x,x}(s,\tau)}\sqrt{\underset{W}{\leftrightarrow}^{Z1,Z1}(s,\tau)}\sqrt{\underset{W}{\leftrightarrow}^{Z1,Z1}(s,\tau)}} \right|^2}{\left(1 - R_{y,Z1}^2(s,\tau)\right)\left(1 - R_{x,Z1}^2(s,\tau)\right)}$$

$$= \frac{\left| \dfrac{\underset{W}{\leftrightarrow}^{y,x}(s,\tau)}{\sqrt{\underset{W}{\leftrightarrow}^{y,y}(s,\tau)}\sqrt{\underset{W}{\leftrightarrow}^{x,x}(s,\tau)}} - \dfrac{\underset{W}{\leftrightarrow}^{y,Z1}(s,\tau)}{\sqrt{\underset{W}{\leftrightarrow}^{y,y}(s,\tau)}\sqrt{\underset{W}{\leftrightarrow}^{Z1,Z1}(s,\tau)}} \cdot \dfrac{\overline{\underset{W}{\leftrightarrow}^{x,Z1}(s,\tau)}}{\sqrt{\underset{W}{\leftrightarrow}^{x,x}(s,\tau)}\sqrt{\underset{W}{\leftrightarrow}^{Z1,Z1}(s,\tau)}} \right|^2}{\left(1 - R_{y,Z1}^2(s,\tau)\right)\left(1 - R_{x,Z1}^2(s,\tau)\right)}$$

$$= \frac{\left| \gamma_{y,x}(s,\tau) - \gamma_{y,Z1}(s,\tau)\overline{\gamma_{x,Z1}(s,\tau)} \right|^2}{\left(1 - R_{y,Z1}^2(s,\tau)\right)\left(1 - R_{x,Z1}^2(s,\tau)\right)} \tag{RC2}$$

Namely, when only one variable (e.g., $Z1$) is excluded, Eq.(9) can be written as

$$\rho_{y,x\cdot Z1}^2 = \frac{\left| \gamma_{y,x}(s,\tau) - \gamma_{y,Z1}(s,\tau)\overline{\gamma_{x,Z1}(s,\tau)} \right|^2}{\left(1 - R_{y,Z1}^2(s,\tau)\right)\left(1 - R_{x,Z1}^2(s,\tau)\right)} \tag{RC3}$$

Eq. (RC3) will be added to revision as Eq. (14), and the derivation process for this equation will be added to the supplementary.

In the case of one excluding variable ($Z = \{Z_1\}$), Mihanović et al. (2009) suggested that the PWC can be calculated by an equation analogous to the traditional partial correlation squared (Kenney and Keeping, 1939). Their equation is the same to Eq. (14 or RC3). Unfortunately, Ng and Chan (2012a) might have misinterpreted the equation of Mihanović et al. (2009) and developed Matlab code for calculating PWC using the equation expressed as

$$\rho_{y,x\cdot Z1}^2 = \frac{\left| R_{y,x}(s,\tau) - R_{y,Z1}(s,\tau)\,R_{x,Z1}(s,\tau) \right|^2}{\left(1 - R_{y,Z1}^2(s,\tau)\right)\left(1 - R_{x,Z1}^2(s,\tau)\right)} \tag{RC4 (or 15 in the revision)}$$

where $R_{y,x}(s,\tau)$, $R_{y,Z1}(s,\tau)$, and $R_{x,Z1}(s,\tau)$ are the square root of $R_{y,x}^2(s,\tau)$, $R_{y,Z1}^2(s,\tau)$, $R_{x,Z1}^2(s,\tau)$, respectively. $R_{y,Z1}^2(s,\tau)$ and $R_{x,Z1}^2(s,\tau)$ can be calculated from Eq. (10) by replacing $y$ and $x$ with their corresponding variables. Eq. (15) has been widely used to calculate PWC in case of one excluding variable (Aloui et al., 2018; Altarturi et al., 2018; Jia et al., 2018; Li et al., 2018; Mutascu and Sokic, 2020; Ng and Chan, 2012b; Rathinasamy et al., 2017; Wu et al., 2020).

Note that complex coherence and real coherence are involved in the numerators of Eqs. (14) and (15), respectively, while the denominators are exactly the same. Further comparison indicates that Eq. (RC4) underestimates PWC value relative to Eq. (14) unless $\gamma_{y,x}(s,\tau)$

and $\gamma_{y,Z1}(s,\tau)\,\overline{\gamma_{x,Z1}(s,\tau)}$ in Eq. (14) are collinear (i.e., their arguments are identical) under which the two equations produce the same PWC values. Differences between Eqs. (14) and (15) will be discussed further using both artificial data and real dataset. For the comparison purpose, we refer to Eqs. (14) and (15) as new method and classical method, respectively.

Therefore, the differences in PWC values calculated from the two methods are context-specific. As the Referee #2 mentioned, although the difference between the Mihanovic et al. (2009) model (Eq.15) and the proposed model (Eq.14) are small, i.e., the difference of PWC values is only 0.03 for the artificial data, but Eq.14 produces PWC closer to 1.

In addition, the comparison of these two methods using real data indicated that the difference between the two methods can be large. As an example, Figure RC2 and Figure RC3a shows big differences of PWC values between these two methods at scales of around 12 months (1 year). Mean PWC values by the new method were consistently higher than the classical method, and the differences ranged from 0.4 to 0.6 around the scale of 1 year (Figure RC3b).This highlights that the new method produces more accurate results than the classical method.

In the revision, we will incorporate these discussions either in the main body of the paper or in the supplementary.

[Figure]

Figure RC2. Partial wavelet coherency (PWC) between evaporation (E) and relative humidity (RH) after excluding the effect of mean temperature (T) calculated by the new method. (subplot of Figure 3d in the revision)

[Figure]

Figure RC3. Partial wavelet coherency (PWC) between evaporation (E) and relative humidity (RH) after excluding the effect of mean temperature (T) using the classical method (a) and differences in PWC between the new method and classical method as a function of scale (b).

***Comment #3:***
*(3) Regarding the structure, is it more suitable to reorganize the Section 3 and 4, that is, the artificial data and their results are analyzed and discussed in Section 3, while those of real data are analyzed and discussed in Section 4?*

***Response #3:***

Thanks for the good suggestion on paper structure. In the revision, we will follow the order of data description, data analysis, results and discussion for each of artificial dataset and real data. To reduce the length of this paper, we will take the suggestion from Referee #2 to remove the real data related to soil water content by adding more about the introduction of the wavelet methods and in-depth discussion of the advantages and weaknesses of the new method.

Thanks again for your constructive comment.

**References:**
Aloui, C., Hkiri, B., Hammoudeh, S., Shahbaz, M., 2018. A multiple and partial wavelet analysis of the oil price, inflation, exchange rate, and economic growth nexus in Saudi Arabia. Emerging Markets Finance and Trade 54(4), 935-956.
Altarturi, B.H.M., Alshammari, A.A., Saiti, B., Erol, T., 2018. A three-way analysis of the relationship between the USD value and the prices of oil and gold: A wavelet analysis. Aims Energy 6(3), 487-504.
Grinsted, A., Moore, J.C., Jevrejeva, S., 2004. Application of the cross wavelet transform and wavelet coherence to geophysical time series. Nonlinear Processes in Geophysics 11(5/6), 561-566.
Jia, X., Zha, T., Gong, J., Zhang, Y., Wu, B., Qin, S., Peltola, H., 2018. Multi-scale dynamics and environmental controls on net ecosystem $CO_2$ exchange over a temperate semiarid shrubland. Agricultural and Forest Meteorology 259, 250-259.
Kenney, J.F., Keeping, E.S., 1939. Mayhematics of Statistics. D. van Nostrand.
Li, H., Dai, S., Ouyang, Z., Xie, X., Guo, H., Gu, C., Xiao, X., Ge, Z., Peng, C., Zhao, B., 2018. Multi-scale temporal variation of methane flux and its controls in a subtropical tidal salt marsh in eastern China. Biogeochemistry 137(1-2), 163-179.
Mihanović, H., Orlić, M., Pasarić, Z., 2009. Diurnal thermocline oscillations driven by tidal flow around an island in the Middle Adriatic. Journal of Marine Systems 78, S157-S168.
Mutascu, M., Sokic, A., 2020. Trade openness-$CO_2$ emissions nexus: a wavelet evidence from EU. Environmental Modeling & Assessment 25, 1-18.
Ng, E.K., Chan, J.C., 2012a. Geophysical applications of partial wavelet coherence and multiple wavelet coherence. Journal of Atmospheric and Oceanic Technology 29(12), 1845-1853.
Ng, E.K., Chan, J.C., 2012b. Interannual variations of tropical cyclone activity over the north Indian Ocean. International Journal of Climatology 32(6), 819-830.
Rathinasamy, M., Agarwal, A., Parmar, V., Khosa, R., Bairwa, A., 2017. Partial wavelet coherence analysis for understanding the standalone relationship between Indian Precipitation and

Teleconnection patterns. arXiv preprint arXiv:1702.06568.

Si, B.C., Farrell, R.E., 2004. Scale-dependent relationship between wheat yield and topographic indices: A wavelet approach. Soil Sci Soc Am J 68(2), 577-587.

Wendroth, O., Alomran, A.M., Kirda, C., Reichardt, K., Nielsen, D.R., 1992. State-Space Approach to Spatial Variability of Crop Yield. Soil Sci Soc Am J 56(3), 801-807.

Wu, K., Zhu, J., Xu, M., Yang, L., 2020. Can crude oil drive the co-movement in the international stock market? Evidence from partial wavelet coherence analysis. The North American Journal of Economics and Finance, 101194.

---

## Author Comment (AC2) · 7 Oct 2020

**Response to Anonymous Referee #2**

*Comment #1:*
**Summary**  *In this technical note, the authors propose a method for identifying relationships between two variables for the case where the two variables are correlated to other variables themselves. They apply their updated partial wavelet coherency' (PWC) method to a synthetic dataset and two real-world applications and show that this updated PWC model shows similar performance as existing PWC models. They conclude that their model outperforms existing models because it provides phase information and allows for excluding several correlated variables from the PWC.*

*Response  #1:*

Many thanks for your comment. We think the new method outperforms the existing one from the three aspects: (1) more accurate results because of the theoretical differences (will be explained below); (2) inclusion of phase information; and (3) any number of excluding variables can be considered.

Below we will respond to each of your comments.

*Comment #2:*

**General remarks**  *I think that the study addresses a question of interest to the hydrological community, i.e. 'how can we identify the most important driving variables of a certain phenomena at different time scales'. The technical note is generally well structured. However, I think that it lacks a didactical and detailed introduction to the topic, problem, and wavelet analysis. The introduction would significantly benefit from providing examples of when the identification of bivariate relationships are important (i.e. providing a motivation for the study), an in-depth introduction to wavelet analysis (for the readers who are not yet too familiar with the topic), and an introduction to the terminology used. Extending the introduction will increase the length of the note and I suggest removing the practical example number 2 instead. I think it does not provide additional insights regarding the performance of the method proposed compared to the statements that were already made based on the synthetic data and the first practical example. Since the new method does not seem to clearly outperform existing methods, I would better explain why adding phase information and excluding several confounding variables is beneficial for the analysis. I would also add a more detailed discussion of model weaknesses, especially the implications of detecting spurious correlations. In addition, the note would profit from careful language editing.*

*Response  #2:*

More detailed information on the general wavelet analysis, PWC, and problem of existing methods will be added in the Introduction and Theory sections (see more details below). The importance of bivariate relationships will be explained by adding "The BWC partitions correlation between two variables into different locations and scales, which are different from the overall relationships at the sampling scale as shown from the traditional correlation coefficient. For example, BWC indicated that soil water content (SWC) of a hummocky landscape in the Canadian Prairies was negatively correlated to soil organic carbon (SOC) content at a slope scale (50 m), but was positively correlated to SOC at a watershed scale (120 m) in summer due to the different processes involved at different scales (Hu et al., 2017). Because the positive correlation cancels with the negative at different scales and/ or locations, the traditional correlation coefficient between SWC and SOC is absent, which is misleading." In terms of wavelet analysis, we will add "The wavelet analyses are based on wavelet transform using mother wavelet function which expands spatial (or

time) series into location-scale (or time-frequency) space for identification of localized intermittent scales (or periodicities)."

The original motivation to have both real datasets is to demonstrate that the proposed method can be used for both spatial and temporal data. We agree that more detailed introduction will increase the length of the paper, so we will remove the results related to SWC. Its suitability to both spatial and temporal data will be mentioned in the revision.

The new method and the existing method have theoretical differences. We will derive the equation for calculating PWC in case of one excluding variable from our Eq. (9). After we carefully check the paper of Mihanovic et al. (2009), it looks like their equation is the same to our equation in case of one excluding variables. Unfortunately, Ng and Chan (2012a) might have misinterpreted the equation of Mihanović et al. (2009) and developed Matlab code for calculating PWC by replacing complex coherence with the real coherence between two variables (please see the detail below). Unfortunately, the code of Ng and Chan (2012a) has been widely used. Not surprisingly, the equation of Ng and Chan (2012a) usually underestimates PWC value relative to the new method. Analysis from the real data has indicated that the differences between two methods can be large. The related discussion will be added either in the main body of the paper or in the supplementary. Meanwhile, more discussion on why adding phase information and excluding several confounding variables is beneficial for the analysis will be added.

A separate discussion section will be added by including a more detailed discussion of model advantages (e.g., the three aspects mentioned in the Response #1) and weaknesses (including spurious correlations and multiple-testing).

Language has been carefully checked by editors from our publication office.

Please see the details below on how we will address the comments you have made.

**Major points**

***Comment #3:***
*1. **Abstract:** The abstract is not very accessible to non-wavelet-specialists. I would provide a short example for when such an analysis would be necessary/beneficial and shortly summarize what wavelet coherence analysis is all about. Please also shortly explain why PWC has been introduced in the first place (l. 12). I would also mention the datasets used for model evaluations (l. 14). I think the statement 'producing more accurate results' (l. 18) needs justification, otherwise it is not very credible. I would exclude lines 21-24 because this is a technical note and specific results regarding the example applications going beyond model performance are in my opinion not of interest here.*
***Response #3:***

We will add "It is a measure of correlation between two spatial (or time) series in the location-scale (time-frequency) domain. This method is particularly suited to geoscience where relationships between multiple variables commonly differ with locations or/and scales due to varying processes involved across different scales and locations." to explain what is wavelet coherence and when it would benefit.
The PWC was introduced "to detect the scale-specific and localized bivariate relationships by excluding the effects of other variables".

The description of dataset used for model evaluations will be "Both stationary and non-stationary artificial datasets with response variable being the sum of five cosine waves at 256 locations are used for method tests.".

Why the new method produces more accurate results will be explained by adding "Compared with the previous PWC calculation, the new method produces more accurate results in case of one excluding variable because bivariate real coherence rather than the bivariate complex coherence was mistakenly used in the previous PWC calculation".

Lines 21-24 from previous submission have been removed.

***Comment #4:***

***2. Introduction:*** *The introduction should in my opinion provide a motivation for the use of PWC methods, also for non-specialists on the topic e.g. by providing examples of important bivariate relationships in the geosciences and why we may be interested in them. In addition, an introduction to wavelet analysis in general and wavelet coherence analysis in particular should be provided. The reader should also be made familiar with the terminology used, e.g. what kind of scales are you talking about and what is an 'excluding variable'. A clear motivation for why excluding variables and including phases matters is required to underline the benefits of the methods later on in the results and conclusions sections (l. 57-58). Currently, the introduction does not very well prepare readers for what they are going to read in the methods and results sections.*

***Response #4:***

The important bivariate relationships in geosciences will be explained by adding "The BWC gives correlation between two variables at different locations and scales rather than the overall relationships at the sampling scale as obtained from the traditional correlation coefficient. For example, BWC indicated that soil water content (SWC) of a hummocky landscape in the Canadian Prairies was negatively correlated to soil organic carbon (SOC) content at a slope scale (50 m), but was positively correlated to SOC at a watershed scale (120 m) in summer due to the different processes involved at different scales (Hu et al., 2017). Because of the different correlations at different scales, the traditional correlation coefficient indicated that SWC was not correlated to SOC, which is misleading..".

The motivation for the use of PWC method is further explained by adding "Partial correlation analysis is one such method to avoid the misleading relationships resulting from the interdependence between other variables and both predictor and response variables (Kenney and Keeping, 1939)" and "For example, PWC analysis indicated that Southern Oscillation Index and Pacific Decadal Oscillation did not affect precipitation across India, while this was misinterpreted by the BWC analysis because of their interdependence on Niño 3.4 that affects precipitation (Rathinasamy et al., 2017)."

An introduction to wavelet analysis in general will be added as "The wavelet analyses are based on wavelet transform using mother wavelet function which expands spatial (or time) series into location-scale (or time-frequency) space for identification of localized intermittent scales (or periodicities)."

When we talk about scale, it can mean spatial or temporal scale depending on if the dataset are spatial series or time series. To avoid repeatedly addressing if this is related to spatial or time scale, we will define it at the first time by adding "For convenience, we will mainly refer to location and scale irrespective of spatial or temporal series unless otherwise mentioned".

Excluding variable refers to "variable whose influence on the response variable is excluded".

The explanation on the motivation for why excluding variables and including phases matter will be added as "The coherence between response and predictor variables can still be misleading if more than one variable is interdependent with the predictor variable. This is especially true if these variables are correlated with the predictor variable at different locations or/and scales. In addition, the types of correlation (i.e., positive or negative) especially at different locations and scales remains unclear without phase information.".in the introduction.

***Comment #5:***

***3. Theory:*** *I think that you should start even simpler here and provide a short introduction to wavelet analysis (difference between discrete and complex, terminology) and wavelet coherence analysis. In addition, it is unclear to me what exactly the difference between classical PWC and your proposed method is (l. 74-76). Currently, it is not entirely clear to me how the Monte Carlo experiment was performed (l. 108-110). Could you please slightly expand this section?*

***Response #5:***

We will add the introduction to wavelet analysis, wavelet coherence analysis and associated equations at start of the Theory section. Here we assume you mean difference between discrete wavelet transform and continuous wavelet transform. The addition will be like:

"Wavelet analysis is based on the calculations of wavelet coefficients using wavelet transform at different scales and locations for each variable involved. Two types of wavelet transform exist including continuous wavelet transform (CWT) and discrete wavelet transform (DWT). While the DWT is mainly used for data compression and noise reduction, the CWT is widely used for extracting scale-specific and localized features as is the case of this study (Grinsted et al., 2004). For the CWT, the Morlet wavelet is used as a mother wavelet function to transform a spatial (or time) series into location-scale (or time-frequency) domain which allows us to identify both location-specific amplitude and phase information of wavelet coefficients at different scales (Torrence and Compo, 1998). From wavelet coefficients, auto- and cross-wavelet power spectra for two variables can be calculated as the product of wavelet coefficient and complex conjugate of itself (auto-wavelet power spectra) or another variable (cross-wavelet power spectra). The BWC is calculated as the ratio of smoothed cross-wavelet power spectra to the product of two auto-wavelet power spectra (Grinsted et al., 2004). Hu and Si (2016) extended wavelet coherence from two to multiple ($\geq 3$) variables and developed MWC. Detailed information on the calculations of wavelet coefficients, auto- and cross-wavelet power spectra, BWC, and MWC based on the CWT can be found elsewhere (Torrence and Compo, 1998; Grinsted et al., 2004; Si and Farrell, 2004; Si, 2008; Hu and Si, 2016; Hu et al., 2017). Here, we will only introduce the theory and calculation very relevant to PWC. "

In addition, the derivation of Eq.(1) in the original submission from equations of complex partial spectrum in frequency domain and bivariate complex coherence from time-frequency domain will be demonstrated in the supplement as below:

"**S1 Derivation of the complex PWC Eq.(1)**

Complex partial spectrum from frequency (scale)domain (Makhtar et al., 2014) can be used to define that of time-frequency (location-scale) domain, $\overset{\leftrightarrow}{W}^{y,x\cdot Z}(s,\tau)$, which is expressed as

$$\overset{\leftrightarrow}{W}{}^{y,x\cdot Z}(s,\tau) = \overset{\leftrightarrow}{W}{}^{y,x}(s,\tau) - \frac{\overset{\leftrightarrow}{W}{}^{y,Z(s,\tau)}\overline{\overset{\leftrightarrow}{W}{}^{x,Z(s,\tau)}}}{\overset{\leftrightarrow}{W}{}^{Z,Z(s,\tau)}} \tag{S1}$$

where $\overset{\leftrightarrow}{W}$ is the smoothed cross spectrum, $\overline{(\cdot)}$ is the complex conjugate operator, $y$, $x$, and $Z$ ($Z = \{Z_1, Z_2, \cdots, Z_q\}$) refer to the response variable, predictor variable, and excluding variables, respectively. $s$ and $\tau$ refer to scale (frequency) and location (time), respectively.

Given the definition of coherence between two variables $y$ and $x$, their complex coherence $\gamma_{y,x}(s,\tau)$ (Eq.(5)) can be re-written as

$$\gamma_{y,x}(s,\tau) = \frac{\overset{\leftrightarrow}{W}{}^{y,x(s,\tau)}}{\sqrt{\overset{\leftrightarrow}{W}{}^{y,y(s,\tau)}\overset{\leftrightarrow}{W}{}^{x,x(s,\tau)}}} \tag{S2}$$

Then we can define complex partial coherence as

$$\gamma_{y,x\cdot Z}(s,\tau) = \frac{\overset{\leftrightarrow}{W}{}^{y,x\cdot Z(s,\tau)}}{\sqrt{\overset{\leftrightarrow}{W}{}^{y,y\cdot Z(s,\tau)}\overset{\leftrightarrow}{W}{}^{x,x\cdot Z(s,\tau)}}} \tag{S3}$$

Based on Eq. (S1) and Eqs 2, 3, and 4 ($R^2_{y,x\cdot Z}(s,\tau) = \frac{\overset{\leftrightarrow}{W}{}^{y,Z(s,\tau)}\overset{\leftrightarrow}{W}{}^{Z,Z(s,\tau)-1}\overline{\overset{\leftrightarrow}{W}{}^{x,Z(s,\tau)}}}{\overset{\leftrightarrow}{W}{}^{y,x(s,\tau)}}$ ,

$R^2_{y,Z}(s,\tau) = \frac{\overset{\leftrightarrow}{W}{}^{y,Z(s,\tau)}\overset{\leftrightarrow}{W}{}^{Z,Z(s,\tau)-1}\overline{\overset{\leftrightarrow}{W}{}^{y,Z(s,\tau)}}}{\overset{\leftrightarrow}{W}{}^{y,y(s,\tau)}}$ , and $R^2_{x,Z}(s,\tau) = \frac{\overset{\leftrightarrow}{W}{}^{x,Z(s,\tau)}\overset{\leftrightarrow}{W}{}^{Z,Z(s,\tau)-1}\overline{\overset{\leftrightarrow}{W}{}^{x,Z(s,\tau)}}}{\overset{\leftrightarrow}{W}{}^{x,x(s,\tau)}}$ )

we obtain

$$\overset{\leftrightarrow}{W}{}^{y,x\cdot Z}(s,\tau) = \overset{\leftrightarrow}{W}{}^{y,x}(s,\tau)\left(1 - \frac{\overset{\leftrightarrow}{W}{}^{y,Z(s,\tau)}\overline{\overset{\leftrightarrow}{W}{}^{x,Z(s,\tau)}}}{\overset{\leftrightarrow}{W}{}^{Z,Z(s,\tau)}\overset{\leftrightarrow}{W}{}^{y,x(s,\tau)}}\right) = \overset{\leftrightarrow}{W}{}^{y,x}(s,\tau)\left(1 - R^2_{y,x\cdot Z}(s,\tau)\right) \tag{S4}$$

$$\overset{\leftrightarrow}{W}{}^{y,y\cdot Z}(s,\tau) = \overset{\leftrightarrow}{W}{}^{y,y}(s,\tau)\left(1 - \frac{\overset{\leftrightarrow}{W}{}^{y,Z(s,\tau)}\overline{\overset{\leftrightarrow}{W}{}^{y,Z(s,\tau)}}}{\overset{\leftrightarrow}{W}{}^{Z,Z(s,\tau)}\overset{\leftrightarrow}{W}{}^{y,y(s,\tau)}}\right) = \overset{\leftrightarrow}{W}{}^{y,y}(s,\tau)\left(1 - R^2_{y,Z}(s,\tau)\right) \tag{S5}$$

$$\overset{\leftrightarrow}{W}{}^{x,x\cdot Z}(s,\tau) = \overset{\leftrightarrow}{W}{}^{x,x}(s,\tau)\left(1 - \frac{\overset{\leftrightarrow}{W}{}^{x,z(s,\tau)}\overline{\overset{\leftrightarrow}{W}{}^{x,Z(s,\tau)}}}{\overset{\leftrightarrow}{W}{}^{Z,Z(s,\tau)}\overset{\leftrightarrow}{W}{}^{x,x(s,\tau)}}\right) = \overset{\leftrightarrow}{W}{}^{x,x}(s,\tau)\left(1 - R^2_{x\cdot Z}(s,\tau)\right) \tag{S6}$$

Inserting Eqs S4, S5, and S6 into Eq. (S3), we have

$$\gamma_{y,x\cdot Z}(s,\tau) = \frac{\overset{\leftrightarrow}{W}{}^{y,x(s,\tau)}\left(1-R^2_{y,x\cdot Z}(s,\tau)\right)}{\sqrt{\overset{\leftrightarrow}{W}{}^{y,y(s,\tau)}\left(1-R^2_{y,Z}(s,\tau)\right)\overset{\leftrightarrow}{W}{}^{x,x(s,\tau)}\left(1-R^2_{x\cdot Z}(s,\tau)\right)}} = \frac{\overset{\leftrightarrow}{W}{}^{y,x(s,\tau)}\left(1-R^2_{y,x\cdot Z}(s,\tau)\right)}{\sqrt{\overset{\leftrightarrow}{W}{}^{y,y(s,\tau)}\overset{\leftrightarrow}{W}{}^{x,x(s,\tau)}}\sqrt{\left(1-R^2_{y,Z}(s,\tau)\right)\left(1-R^2_{x\cdot Z}(s,\tau)\right)}} =$$

$$\frac{\left(1-R^2_{y,x\cdot Z}(s,\tau)\right)\gamma_{y,x}(s,\tau)}{\sqrt{\left(1-R^2_{y,Z}(s,\tau)\right)\left(1-R^2_{x\cdot Z}(s,\tau)\right)}} \tag{S7}$$

Obviously, Eq. (S7) and Eq. (1) are identical."

The difference between Eq. (9) and Eq. (14) will be explained by derivation of PWC in the case of one excluding variable from Eq. (1) (This is also addressed in the Response #2 to the referee #1).

So, when only one variable (e.g., $Z1$) is excluded, Eq.(9) ($\rho^2_{y,x\cdot Z} = \frac{\left|1-R^2_{y,x\cdot Z}(s,\tau)\right|^2 R^2_{y,x}(s,\tau)}{\left(1-R^2_{y,Z}(s,\tau)\right)\left(1-R^2_{x,Z}(s,\tau)\right)}$ )

can be written as

$$\rho_{y,x\cdot Z1}^2 = \frac{\left|1-R_{y,x\cdot Z1}^2(s,\tau)\right|^2 R_{y,x}^2(s,\tau)}{\left(1-R_{y,Z1}^2(s,\tau)\right)\left(1-R_{x,Z1}^2(s,\tau)\right)} \tag{RC1}$$

Based on equations (2) in our paper,

$$\rho_{y,x\cdot Z1}^2 = \frac{\left|1-\dfrac{\overset{\leftrightarrow y,Z1}{W}(s,\tau)\ \overset{\leftrightarrow Z1,Z1}{W}(s,\tau)^{-1}\ \overline{\overset{\leftrightarrow x,Z1}{W}(s,\tau)}}{\overset{\leftrightarrow y,x}{W}(s,\tau)}\right|^2 \dfrac{\left|\overset{\leftrightarrow y,x}{W}(s,\tau)\right|^2}{\overset{\leftrightarrow y,y}{W}(s,\tau)\ \overset{\leftrightarrow x,x}{W}(s,\tau)}}{\left(1-R_{y,Z1}^2(s,\tau)\right)\left(1-R_{x,Z1}^2(s,\tau)\right)}$$

$$= \frac{\left|\overset{\leftrightarrow y,x}{W}(s,\tau)-\dfrac{\overset{\leftrightarrow y,Z1}{W}(s,\tau)\overline{\overset{\leftrightarrow x,Z1}{W}(s,\tau)}}{\overset{\leftrightarrow Z1,Z1}{W}(s,\tau)}\right|^2}{\overset{\leftrightarrow y,y}{W}(s,\tau)\overset{\leftrightarrow x,x}{W}(s,\tau)\left(1-R_{y,Z1}^2(s,\tau)\right)\left(1-R_{x,Z1}^2(s,\tau)\right)}$$

$$= \frac{\dfrac{1}{\sqrt{\left(\overset{\leftrightarrow y,y}{W}(s,\tau)\overset{\leftrightarrow x,x}{W}(s,\tau)\right)^2}}\left|\overset{\leftrightarrow y,x}{W}(s,\tau)-\dfrac{\overset{\leftrightarrow y,Z1}{W}(s,\tau)\overline{\overset{\leftrightarrow x,Z1}{W}(s,\tau)}}{\left(\sqrt{\overset{\leftrightarrow Z1,Z1}{W}(s,\tau)}\right)^2}\right|^2}{\left(1-R_{y,Z1}^2(s,\tau)\right)\left(1-R_{x,Z1}^2(s,\tau)\right)}$$

$$= \frac{\left|\dfrac{\overset{\leftrightarrow y,x}{W}(s,\tau)}{\sqrt{\overset{\leftrightarrow y,y}{W}(s,\tau)}\sqrt{\overset{\leftrightarrow x,x}{W}(s,\tau)}}-\dfrac{\overset{\leftrightarrow y,Z1}{W}(s,\tau)\ \overline{\overset{\leftrightarrow x,Z1}{W}(s,\tau)}}{\sqrt{\overset{\leftrightarrow y,y}{W}(s,\tau)}\sqrt{\overset{\leftrightarrow x,x}{W}(s,\tau)}\sqrt{\overset{\leftrightarrow Z1,Z1}{W}(s,\tau)}\sqrt{\overset{\leftrightarrow Z1,Z1}{W}(s,\tau)}}\right|^2}{\left(1-R_{y,Z1}^2(s,\tau)\right)\left(1-R_{x,Z1}^2(s,\tau)\right)}$$

$$= \frac{\left|\dfrac{\overset{\leftrightarrow y,x}{W}(s,\tau)}{\sqrt{\overset{\leftrightarrow y,y}{W}(s,\tau)}\sqrt{\overset{\leftrightarrow x,x}{W}(s,\tau)}}-\dfrac{\overset{\leftrightarrow y,Z1}{W}(s,\tau)}{\sqrt{\overset{\leftrightarrow y,y}{W}(s,\tau)}\sqrt{\overset{\leftrightarrow Z1,Z1}{W}(s,\tau)}}\cdot\dfrac{\overline{\overset{\leftrightarrow x,Z1}{W}(s,\tau)}}{\sqrt{\overset{\leftrightarrow x,x}{W}(s,\tau)}\sqrt{\overset{\leftrightarrow Z1,Z1}{W}(s,\tau)}}\right|^2}{\left(1-R_{y,Z1}^2(s,\tau)\right)\left(1-R_{x,Z1}^2(s,\tau)\right)}$$

$$= \frac{\left|\gamma_{y,x}(s,\tau)-\gamma_{y,Z1}(s,\tau)\overline{\gamma_{x,Z1}(s,\tau)}\right|^2}{\left(1-R_{y,Z1}^2(s,\tau)\right)\left(1-R_{x,Z1}^2(s,\tau)\right)} \tag{RC2}$$

Namely, when only one variable (e.g., $Z1$) is excluded, Eq.(9) can be written as

$$\rho_{y,x\cdot Z1}^2 = \frac{\left|\gamma_{y,x}(s,\tau)-\gamma_{y,Z1}(s,\tau)\overline{\gamma_{x,Z1}(s,\tau)}\right|^2}{\left(1-R_{y,Z1}^2(s,\tau)\right)\left(1-R_{x,Z1}^2(s,\tau)\right)} \tag{RC3}$$

Eq. (RC3) will be added to revision as Eq. (14), and the derivation process for be added to the supplementary.

In the case of one excluding variable ($Z = \{Z_1\}$), Mihanović et al. (2009) suggested that the PWC can be calculated by an equation analogous to the traditional partial correlation squared (Kenney and Keeping, 1939). Their equation is the same to Eq. (14 or RC3). Unfortunately, Ng and Chan (2012a) might have misinterpreted the equation of Mihanović et al. (2009) and developed Matlab code for calculating PWC using the equation expressed as

$$\rho^2_{y,x\cdot Z1} = \frac{\left|R_{y,x}(s,\tau) - R_{y,Z1}(s,\tau)\,R_{x,Z1}(s,\tau)\right|^2}{\left(1 - R^2_{y,Z1}(s,\tau)\right)\left(1 - R^2_{x,Z1}(s,\tau)\right)} \qquad \text{(RC4) (or 15 in the revision)}$$

where $R_{y,x}(s,\tau)$, $R_{y,Z1}(s,\tau)$, and $R_{x,Z1}(s,\tau)$ are the square root of $R^2_{y,x}(s,\tau)$, $R^2_{y,Z1}(s,\tau)$, $R^2_{x,Z1}(s,\tau)$, respectively. $R^2_{y,Z1}(s,\tau)$ and $R^2_{x,Z1}(s,\tau)$ can be calculated from Eq. (10) by replacing $y$ and $x$ with their corresponding variables. Eq. (15) has been widely used to calculate PWC in case of one excluding variable (Aloui et al., 2018; Altarturi et al., 2018; Jia et al., 2018; Li et al., 2018; Mutascu and Sokic, 2020; Ng and Chan, 2012b; Rathinasamy et al., 2017; Wu et al., 2020). Note that complex coherence and real coherence are involved in the numerators of Eqs. (14) and (15), respectively, while the denominators are exactly the same. Further comparison indicates that Eq. (RC4) underestimates PWC value relative to Eq. (14) unless unless $\gamma_{y,x}(s,\tau)$ and $\gamma_{y,Z1}(s,\tau)\,\overline{\gamma_{x,Z1}(s,\tau)}$ in Eq. (14) are collinear (i.e., their arguments are identical) under which the two equations produce the same PWC values. Differences between Eqs. (14) and (15) will be discussed further using both artificial data and real dataset. For the comparison purpose, we refer to Eqs. (14) and (15) as new method and classical method, respectively.

Therefore, the differences in PWC values calculated from the two methods are context-specific. As the Referee #2 mentioned, although the difference between the Mihanovic et al. (2009) model (Eq.15) and the proposed model (Eq.14) are small, i.e., the difference of PWC values is only 0.03 for the artificial data, but Eq.14 produces PWC closer to 1.

In addition, the comparison of these two methods using real data indicated that the difference between the two methods can be large. As an example, Figure RC2 and Figure RC3a shows big differences of PWC values between these two methods at scales of around 12 month (1 year). Mean PWC values by the new method were consistently higher than the classical method, and the differences ranged from 0.4 to 0.6 around the scale of 1 year (Figure RC3b).This highlights that the new method produces more accurate results than the classical method.

In the revision copy, we will incorporate these discussions either in the main body of the paper or in the supplementary.

[Figure]

Figure RC2. Partial wavelet coherency (PWC) between evaporation (E) and relative humidity (RH) after excluding the effect of mean temperature (T) calculated by the new method. (subplot of Figure 3d in the revision)

[Figure]

Figure RC3. Partial wavelet coherency (PWC) between evaporation (E) and relative humidity (RH) after excluding the effect of mean temperature (T) using the classical method (a) and differences in PWC between the new method and classical method as a function of scale (b).

Monte Carlo method is explained in more details by adding why we chose AR1 model and how many repeats are needed as "The first-order autoregressive model (AR(1)) is chosen because it can be used to simulate most geoscience data very well (Wendroth et al., 1992; Grinsted et al., 2004; Si and Farrell, 2004). Different combinations of r1 values (i.e., 0.0, 0.5, and 0.9) were used to generate 10 to 10 000 AR(1) series with three, four and five variables. Our results indicate that the noise combination has little impact on the PWC values at the 95% confidence level as also found by Grinsted et al. (2004) for the BWC case (data not shown). The relative difference of PWC at the 95% confidence level compared to that calculated from 10 000 AR(1) series decreases with increase in number of AR(1) series. When the number of AR(1) is above 300, very low maximum relative difference (e.g., <2%) is observed (Fig. S1 of Sect. S3 in the Supplement). Therefore, repeating number of 300 seems to be efficient for significance test. If calculation time is not a barrier, however, bigger repeating number such as ≥1000 is recommended.".

[Figure]

**Figure S1**. Relationship between maximum relative difference (%) of PWC compared to that calculated from 10 000 AR(1) series (surrogate dataset) versus the number of AR(1) series during the significance test using the Monte Carlo test. Number of scales per octave is 12.

***Comment #6:***

***4. Data and analysis:*** *I would recommend removing the 'soil water content' example (section 4.2.2) because as I can see it does not show anything that has not yet been shown by the 'free evaporation example' in terms of the validity of the model. I would rather invest the space in extending the introduction as outlined in more details above. In the figure captions, I would add a reference to the dataset used to generate it. In addition, I am not sure what you would like to show with the cases where the variable of interest is excluded. I would therefore exclude the results referring to this exercise (e.g. Figure 1 last row and see l. 236-237). I also think that the figures would profit a lot from using labels for subfigures, which would facilitate orientation. To me, the difference between the Mihanovic et al. (2009) model and the proposed model are not evident by looking at the Figures presented (a difference of 0.03 does not seem to be a lot, l.293). Therefore, I think the actual advantages of using this new method should better be worked out and explained before a statement such as 'the new method outperforms the Mihanovic et al. method' (l. 293-294) is made. Please also explain why the inclusion of 'phase information' is an advantage of the new method (l. 312-313).*

***Response #6:***
Thanks for this advice. We removed the soil water content example.

Reference to the dataset used to generate the figures will be added in the figure caption as "All variables were generated by following Yan and Gao (2007) and Hu and Si (2016) and are explained in Section 3.1 and are shown in Fig. S2 of Sect. Ss in the Supplement."

The purpose of showing the cases of variable of interest being excluded is to basically show that the PWC values should be theoretically zero in that case. As we have the similar results in the case of two excluding variables (Figure 3 in the original submission), we will remove this in Figure 1.

We will add a label for each subfigure in the revision.

As we explained below, some theoretical differences exist between these two methods. We will add this into the Theory section as shown above.

In the new discussion section, we will highlight the advantages and weakness of the new method (Please see the details in the Response #7 below).

***Comment #7:***

***5. A proper discussion section is missing:*** *I would add an in-depth discussion of the weaknesses and benefits of the approach and put the new method into perspective by comparing it to existing methods.*

***Response #7:***
Advantages and weaknesses of the method will be added as:

[revised manuscript text omitted]

***Comment #8:***

**6. Conclusions:** *Given the evidence provided in the results section, statements such as 'the new method produces slightly more accurate coherence' do not seem to be justified. As mentioned earlier the benefits of including phase information and excluding several variables need to be better explained. Some of the material presented in this section could be moved to the new discussion section.*

***Response #8:***
As we replied above, we think 'the new method produces more accurate coherence' is justified by considering both the theoretical differences and the example of real data (Figure RC3) above. The benefits of including phase information and excluding several variables will be discussed in the new discussion section as we explained in Response #7.
Yes, a large part from the conclusions part will be moved to the Discussion section as shown in Response #7.

***Comment #9:***

**7. Code availability:** *I would provide the Matlab code via a data/file repository such as HydroShare or Zenodo instead of the supplement (l.27). This would be very helpful for the community and potential users.*

***Response #9:***
We have provided the Matlab code to the figshare (https://figshare.com/s/bc97956f43fe5734c784). Meanwhile, we have also put the updated codes for multiple wavelet coherence (MWC) which is necessary for calculating PWC in the same repository. We have improved the calculation time for MWC.

**Minor points**

***Comment #10:***
*L. 31: please explain what you mean by 'time and space localization'.*
***Response #10:***
We will add an example to show the localization "For example, time series of air temperature usually fluctuates periodically at different scales (e.g., daily and yearly), but abrupt changes (e.g., extremely high or low) in air temperature may occur at a certain instant of time as a result of extreme weather and climate events (e.g., heat and rain).".

***Comment #11:***

*L.34: 'among these methods'*

*Transition from l. 42 to l. 43: very sharp transition from bivariate relationships to prediction. I would try to establish a clear link between the two things.*

***Response #11:***
We will change "Among which" to "Among these wavelet methods".

We're sorry that we are not sure we understood this comment. But we end up with the wide application of multiple wavelet coherence (MWC) method in the previous graph, and the next paragraph we start with what the MWC application has told us. Namely more predictor variables does not necessarily explain more variations in the response variable because predictor variables are usually cross-correlated. Because of the same reason, bivariate relationships can be misleading. Then we call the need to develop partial wavelet coherence (PWC). Now in the revision, we will put them in the same paragraph.

***Comment #12:***

*L. 48: what do you mean by 'this issue'?*

***Response #12:***
We mean "the misleading relationships resulting from the interdependence between other variables and both predictor and response variables".

***Comment #13:***
*L. 50: what kind of scales? Temporal or spatial?*

***Response #13:***
We mean either temporal or spatial scales depending on if the dataset are time series or spatial series. For avoiding repeatedly saying this, we will clarify this at the first time by adding "For convenience, we will mainly refer to location and scale irrespective of spatial or temporal series unless otherwise mentioned.".

***Comment #14:***
*L. 53-54: would combine greenhouse gas emissions and climate in one category.*

***Response #14:***
Actually we mean different things. We mean precipitation by climate, so we will change climate to meteorological science for avoiding confusing.

***Comment #15:***
*L. 61: information 'which will allow to….'*

***Response #15:***
We will add "which will allow to better understand the magnitude and type of bivariate relationships after removing effects from all other interdependent variables"

***Comment #16:***
*L. 61: what do you mean by 'analogy' in this context. I think that rephrasing may be required.*

***Response #16:***
We will change "in analogy with" simply to "from".

***Comment #17:***

*L. 62: Be specific with what you mean by 'it': 'the proposed method'.*

***Response #17:***
We will change it to "The proposed method".

***Comment #18:***
*L. 76: Please explain to the reader what you mean by 'scale' and 'location'.*

***Response #18:***
Scale and location for spatial series correspond to frequency (periodicity) and time, respectively.
"For convenience, we will mainly refer to location and scale irrespective of spatial or temporal series unless otherwise mentioned."

***Comment #19:***
*L. 99: same for 'phase angle'.*

***Response #19:***
We will add its explanation in the bracket as "(i.e., angle between two complex numbers)".

***Comment #20:***
*L. 184-185: can in my opinion be removed.*

***Response #20:***
We will remove this sentence.

***Comment #21:***
*L. 191: what does data refer to? Soil water content?*

***Response #21:***
It refers to soil water datasets. Now removed as you suggested.

***Comment #22:***
*L. 214: 'significance band'.*

***Response #22:***
We will change it to significance band.

***Comment #23:***
*L. 215-216: is this statement underlined by any analysis performed?*

***Response #23:***
Yes. The number is obtained from calculation.

***Comment #24:***
*L. 247: what is the purpose of replacing half of the time series by 0?*

***Response #24:***
As we highlighted in the methodology section, "second half of the original series of y2 (or z2) are replaced by 0 to simulate abrupt changes (i.e., transient and localized feature) of the spatial series".

***Comment #25:***
*L. 261-263: Which feature in the plots actually indicates these 'abrupt changes'?*

***Response #25:***
The abrupt changes were captured by the abrupt transition from coherence of 0 to coherence of 1 as shown in figure 2a and 2e (top 2 at the left hand side of figure 2 in the original submission).

***Comment #26:***
*L. 266: I can only see one wavelet band of high significance in Figure 3. Where is the second one you mention here?*

***Response #26:***
We did not show the results here, but it was shown in Fig. 2 of our previous paper (Hu and Si, 2016). For this reason, the citation of "(Hu and Si, 2016)" will be added here.

***Comment #27:***
*L. 298: introduce term 'octave'.*

***Response #27:***
We will add "octave refers to the scaled distance between two scales with one scale being twice or half of the other."

***Comment #28:***
*L. 363-366: would move this sentence to discussion section.*

***Response #28:***
Yes, we will move this sentence to the discussion section.

Thanks again for your constructive comment.

---

## Author Comment (AC3) · 7 Oct 2020

**Response to Anonymous Referee #3**

Anonymous Referee #3

*Comment #1:*

*In this paper, the authors presented an improved variant of PWC for identifying the relationship between variables. This should be reflected in the title (like Improved PWC etc to be included in the title) to convey novel contribution. Also at present it is misleading like the authors proposes PWC concept.*

*Response #1:*
Many thanks for your comments. We will change the title to "Technical Note: Improved partial wavelet coherency for understanding scale-specific and localized bivariate relationships in geosciences".

*Overall the paper is well written. I recommend for minor revision.*

*Comment #2:*

*Line 18– and producing more accurate results.- pl give quantitative statements*

*Response #2:*
As the two methods in case of one excluding variables have theoretical differences, the outperformance is obvious. However, the degree of outperformance depends, in the case of our artificial dataset, the new method produces PWC values more close to 1 than the existing method as we expect although the difference is not big (e.g., PWC value of 1.0 versus 0.97 between $y$ and $y_2$ at the scale of 8 after excluding the effect of $y_4$). However, the comparison of these two methods using real data indicated that the difference between the two methods can be big. For example, the differences in PWC between evaporation (E) and relative humidity (RH) after excluding the effect of mean temperature (T) can be 0.4-0.6 at the scales of about 1 year. For this reason, rather than giving quantitative statements, we would like to point out why the proposed method produces more accurate results by changing the sentence to "Compared with the previous PWC calculation, the new method produces more accurate results in case of one excluding variable because bivariate real coherence rather than the bivariate complex coherence was used in the previous PWC calculation."

*Comment #3:*
*Line 31- provide the developments in chronological order – should be checked at all places What is the real advantage in bringing the phase information in practical cases? this should be mentioned in the introduction section*

*Response #3:*
All citations will be changed in a chronological order.
The importance of phase information will be explained by adding "the types of correlation (i.e., positive or negative) especially at different locations and scales remains unclear without phase information."

***Comment #4:***

*Line 109 .. sufficient number of times using : : :pl make it clear*
***Response #4:***

As we also replied to RC#1, to address the "sufficient number" issue, different combinations of r1 (first-order autocorrelation coefficient) values (i.e., 0.0, 0.5, and 0.9) were used to generate 10 to 10 000 AR(1) series with three, four and five variables. Our results indicate that the noise combination has little impact on the PWC values at the 95% confidence level as also found by Grinsted et al. (2004) for the BWC case (data not shown). The relative difference of PWC at the 95% confidence level compared to that calculated from 10 000 AR(1) series decreases with increase in number of AR(1) series. When the number of AR(1) is above 300, very low maximum relative difference (e.g., <2%) is observed (Fig. RC1 which will be put in the Supplement as Fig. S1 of Sect. S3). Therefore, repeating number of 300 seems to be efficient for significance test. If calculation time is not a barrier, however, bigger repeating number such as ⩾1000 is recommended. This will be added into the revision.

[Figure]

**Figure RC1**. Relationship between maximum relative difference (%) of PWC compared to that calculated from 10 000 AR(1) series (surrogate dataset) versus the number of AR(1) series during the significance test using the Monte Carlo test. Number of scales per octave is 12.

***Comment #5:***
*Line 214- significance band*
***Response #5:***

We will change it to significance band.

***Comment #6:***
*Conclusion: Avoid the statements like – 'this new method produces slightly more accurate coherence'*
***Response #6:***

As we explained in the Response #2, we will change it to "Compared with the previous PWC method, the new PWC method has the advantage of dealing with more than one excluding variable and providing the phase information associated with the PWC."

***Comment #7:***

*Line 450-455 should be explained better ; how can you overcome such problems ? I think better to provide a discussion section before conclusion where such references and unfamiliar terms can be explained in a better way. Then conclusion section should be presented as more specific*

***Response #7:***

New discussion section will be added by moving this part to the discussion section. In terms of spurious correlations and multiple-testing problem, we will put it to a new section 5.2 weaknesses. Meanwhile, the advantages will be mentioned in section 5.1.

Here will be the changes:

[revised manuscript text omitted]

Thanks again for your constructive comment.

---

## Editor Comment (EC1) · Bettina Schaefli (Editor) · 12 Oct 2020

This technical note received three reviews; all reviewer comments conclude that the paper is interesting for the readers of HESS and give detailed indications on how to improve the paper. This namely includes additional details on some methodological choices. I would like to invite the authors to submit a revised version of the paper that would in particular also greatly benefit from a more didactical introduction (see reviewer 2) and from a carful language check.

---

## Author Response (AR1)

**Response to Editor Dr. Bettina Schaefli**
*Comments to the Author:*
*Dear Authors,*
*I would like to invite you to implement to changes that you discussed in the public discussion. Even if it is a technical note, I suggest you follow the suggestions of reviewer two to make the paper a bit more accesible for non-specialists.*

*Response:*
Many thanks for giving us an opportunity to revise this manuscript. We have revised the paper according to what we presented in the public discussion. We also explain how we revise in the response to each comment below. We have added more information on the wavelet methods both in Introduction and Theory sections as reviewer #2 suggested to make the paper more accessible to general readers. We have also tried to avoid using abbreviations as much as we can.

**Response to Anonymous Referee #1**

Comments from Referee #1

*In this paper, the authors mainly developed a partial wavelet coherency method, for identifying the relationship between variables. It is an important issue but also a difficult problem for geo-data analysis, and the method developed would be helpful for the data analysis in geosciences. The following comments are suggested to be considered for further improving the paper:*

**Comment #1:**
*(1) In lines 108-110: the "sufficient number" should be clarified, as it has a big influence on the uncertainty estimation, that is, what number is sufficient? Furthermore, the reason of using first-order autocorrelation coefficient for MC simulation should be explained and discussed.*
**Response #1:**
Many thanks for your review and positive general comment.
To address the "sufficient number" issue, we added the following sentences "Different combinations of r1 values (i.e., 0.0, 0.5, and 0.9) were used to generate 10 to 10 000 AR(1) series with three, four and five variables. Our results indicate that the noise combination has little impact on the PWC values at the 95% confidence level as also found by Grinsted et al. (2004) for the BWC case (data not shown). The relative difference of PWC at the 95% confidence level compared with that calculated from the 10 000 AR(1) series decreases with the increase in number of AR(1) series. When the number of AR(1) is above 300, a very low maximum relative difference (e.g., <2%) is observed (Fig. S1 of Sect. S3 in the Supplement). Therefore, a repeating number of 300 seems to be sufficient for a significance test. However, if calculation time is not a barrier, a higher repeating number, such as ≥1000, is recommended." at Lines 171-181.

[Figure]

**Figure S1**. Relationship between maximum relative difference (%) of PWC compared to that calculated from 10 000 AR(1) series (surrogate dataset) versus the number of AR(1) series during the significance test using the Monte Carlo test. Number of scales per octave is 12. The first-order autocorrelation coefficients (r1) in brackets refer to those for the response variable (first), predictor variable (second), and excluding variables (third and onwards).

"The first-order autoregressive model (AR(1)) is chosen because it can be used to simulate most geoscience data very well (Wendroth et al., 1992; Grinsted et al., 2004; Si and Farrell, 2004)" (Lines169-171).

*Comment #2:*

*(2) Lines 121-122, some theoretical lines can be provided to show the difference between Eq. (9) and Eq. (14).*

*Response #2:*
The difference between Eq. (9) and Eq. (14) was explained by derivation of PWC in the case of one excluding variable from Eq. (1).

"When only one variable (e.g., Z1) is excluded, Eq.(9) can be written as (see the Supplement (Sect. S2) for the derivation process)

$$\rho_{y,x\cdot Z1}^2 = \frac{\left|\gamma_{y,x}(s,\tau) - \gamma_{y,Z1}(s,\tau)\overline{\gamma_{x,Z1}(s,\tau)}\right|^2}{\left(1 - R_{y,Z1}^2(s,\tau)\right)\left(1 - R_{x,Z1}^2(s,\tau)\right)} \qquad (14) \quad "(Lines\ 163\text{-}165)$$

In the supplementary (Sect. S2), we added the derivation of Eq. (14) from Eq. (9) as follows:

**"S2  Derivation of the PWC in case of one excluding variable (Eq.14) from Eq. (9)**

When only one variable (e.g., Z1) is excluded, Eq.(9) ($\rho_{y,x\cdot Z}^2 = \frac{\left|1 - R_{y,x,Z}^2(s,\tau)\right|^2 R_{y,x}^2(s,\tau)}{\left(1 - R_{y,Z}^2(s,\tau)\right)\left(1 - R_{x,Z}^2(s,\tau)\right)}$ )

can be written as

$$\rho_{y,x\cdot Z1}^2 = \frac{\left|1-R_{y,x,Z1}^2(s,\tau)\right|^2 R_{y,x}^2(s,\tau)}{\left(1-R_{y,Z1}^2(s,\tau)\right)\left(1-R_{x,Z1}^2(s,\tau)\right)} \tag{S8}$$

Based on Eq. (2),

$$\rho_{y,x\cdot Z1}^2 = \frac{\left|1-\frac{\overleftrightarrow{W}^{y,Z1}(s,\tau)\,\overleftrightarrow{W}^{Z1,Z1}(s,\tau)^{-1}\,\overline{\overleftrightarrow{W}^{x,Z1}(s,\tau)}}{\overleftrightarrow{W}^{y,x}(s,\tau)}\right|^2 \frac{\left|\overleftrightarrow{W}^{y,x}(s,\tau)\right|^2}{\overleftrightarrow{W}^{y,y}(s,\tau)\,\overleftrightarrow{W}^{x,x}(s,\tau)}}{\left(1-R_{y,Z1}^2(s,\tau)\right)\left(1-R_{x,Z1}^2(s,\tau)\right)}$$

$$= \frac{\left|\overleftrightarrow{W}^{y,x}(s,\tau)-\frac{\overleftrightarrow{W}^{y,Z1}(s,\tau)\,\overline{\overleftrightarrow{W}^{x,Z1}(s,\tau)}}{\overleftrightarrow{W}^{Z1,Z1}(s,\tau)}\right|^2}{\overleftrightarrow{W}^{y,y}(s,\tau)\,\overleftrightarrow{W}^{x,x}(s,\tau)\left(1-R_{y,Z1}^2(s,\tau)\right)\left(1-R_{x,Z1}^2(s,\tau)\right)}$$

$$= \frac{\frac{1}{\sqrt{\left(\overleftrightarrow{W}^{y,y}(s,\tau)\,\overleftrightarrow{W}^{x,x}(s,\tau)\right)^2}}\left|\overleftrightarrow{W}^{y,x}(s,\tau)-\frac{\overleftrightarrow{W}^{y,Z1}(s,\tau)\,\overline{\overleftrightarrow{W}^{x,Z1}(s,\tau)}}{\left(\sqrt{\overleftrightarrow{W}^{Z1,Z1}(s,\tau)}\right)^2}\right|^2}{\left(1-R_{y,Z1}^2(s,\tau)\right)\left(1-R_{x,Z1}^2(s,\tau)\right)}$$

$$= \frac{\left|\frac{\overleftrightarrow{W}^{y,x}(s,\tau)}{\sqrt{\overleftrightarrow{W}^{y,y}(s,\tau)}\sqrt{\overleftrightarrow{W}^{x,x}(s,\tau)}}-\frac{\overleftrightarrow{W}^{y,Z1}(s,\tau)\,\overline{\overleftrightarrow{W}^{x,Z1}(s,\tau)}}{\sqrt{\overleftrightarrow{W}^{y,y}(s,\tau)}\sqrt{\overleftrightarrow{W}^{x,x}(s,\tau)}\sqrt{\overleftrightarrow{W}^{Z1,Z1}(s,\tau)}\sqrt{\overleftrightarrow{W}^{Z1,Z1}(s,\tau)}}\right|^2}{\left(1-R_{y,Z1}^2(s,\tau)\right)\left(1-R_{x,Z1}^2(s,\tau)\right)}$$

$$= \frac{\left|\frac{\overleftrightarrow{W}^{y,x}(s,\tau)}{\sqrt{\overleftrightarrow{W}^{y,y}(s,\tau)}\sqrt{\overleftrightarrow{W}^{x,x}(s,\tau)}}-\frac{\overleftrightarrow{W}^{y,Z1}(s,\tau)}{\sqrt{\overleftrightarrow{W}^{y,y}(s,\tau)}\sqrt{\overleftrightarrow{W}^{Z1,Z1}(s,\tau)}}\cdot\frac{\overline{\overleftrightarrow{W}^{x,Z1}(s,\tau)}}{\sqrt{\overleftrightarrow{W}^{x,x}(s,\tau)}\sqrt{\overleftrightarrow{W}^{Z1,Z1}(s,\tau)}}\right|^2}{\left(1-R_{y,Z1}^2(s,\tau)\right)\left(1-R_{x,Z1}^2(s,\tau)\right)}$$

$$= \frac{\left|\gamma_{y,x}(s,\tau)-\gamma_{y,Z1}(s,\tau)\overline{\gamma_{x,Z1}(s,\tau)}\right|^2}{\left(1-R_{y,Z1}^2(s,\tau)\right)\left(1-R_{x,Z1}^2(s,\tau)\right)} \tag{S9} \text{''}$$

 Later on, we presented the equation for calculating PWC in the classical method and discussed the theoretical differences between two methods in case of one excluding variable at Lines 185-204.

"In the case of one excluding variable ($Z = \{Z_1\}$), Mihanović et al. (2009) suggested that the PWC can be calculated by an equation analogous to the traditional partial correlation squared (Kenney and Keeping, 1939) without giving the detailed derivation process. Their equation is the same as Eq. (14). Unfortunately, Ng and Chan (2012a) might have misinterpreted the equation of Mihanović et al. (2009) and developed Matlab code for calculating PWC using the equation expressed as

$$\rho_{y,x \cdot Z1}^2 = \frac{\left| R_{y,x}(s,\tau) - R_{y,Z1}(s,\tau) \, R_{x,Z1}(s,\tau) \right|^2}{\left(1 - R_{y,Z1}^2(s,\tau)\right)\left(1 - R_{x,Z1}^2(s,\tau)\right)} \qquad\qquad (15)$$

where $R_{y,x}(s,\tau)$, $R_{y,Z1}(s,\tau)$, and $R_{x,Z1}(s,\tau)$ are the square root of $R_{y,x}^2(s,\tau)$, $R_{y,Z1}^2(s,\tau)$, $R_{x,Z1}^2(s,\tau)$, respectively. $R_{y,Z1}^2(s,\tau)$ and $R_{x,Z1}^2(s,\tau)$ can be calculated from Eq. (10) by replacing $y$ and $x$ with their corresponding variables. Eq. (15) has been widely used to calculate PWC in the case of one excluding variable (Ng and Chan, 2012b; Rathinasamy et al., 2017; Aloui et al., 2018; Altarturi et al., 2018b; Jia et al., 2018; Li et al., 2018; Mutascu and Sokic, 2020; Wu et al., 2020). Note that complex coherence and real coherence are involved in the numerators of Eqs. (14) and (15), respectively, while the denominators are exactly the same. Further comparison indicates that Eq. (15) underestimates PWC value relative to Eq. (14) unless $\gamma_{y,x}(s,\tau)$ and $\gamma_{y,Z1}(s,\tau)\,\overline{\gamma_{x,Z1}(s,\tau)}$ in Eq. (14) are collinear (i.e., their arguments are identical) under which the two equations produce the same PWC values. Differences between Eqs. (14) and (15) will be discussed further using both artificial data and a real dataset. For comparison purposes, we refer to Eqs. (14) and (15) as the new method and the classical method, respectively. "

The differences in PWC values calculated from the two methods (Eq. 14 and 15) are context-specific. As the Referee #2 mentioned, although the difference between the Mihanovic et al. (2009) model (Eq.15) and the proposed model (Eq.14) are small, i.e., the difference of PWC values is only 0.03 for the artificial data, Eq.14 produces PWC closer to 1.

In addition, the comparison of these two methods using real data indicated that the difference between the two methods can be large. As an example, mean PWC values between E and RH after excluding the effects of T by the new method were consistently higher than the classical method, and the differences ranged from 0.4 to 0.6 around the scale of 1 year. This highlights that the new method produces more accurate results than the classical method.

These have been added to the discussion section at Lines 414-438 as:

"The differences between the new method (Eq.14) and the classical method (Eq. 15) are compared using both the artificial and real datasets. Except for the phase information, the two methods generally produce comparable coherence for the artificial dataset for the case of one excluding variable (Fig. S5 of Sect. S3 in the Supplement). However, the new PWC method produces consistently and slightly higher coherence than the classical method. For example, their mean PWCs between $y$ and $y_2$ at the scale of 8 after excluding the effect of $y_4$ are 1.00 and 0.97, respectively. This indicates that the new method produces coherence between $y$ and $y_2$ at the scale (8) of $y_2$ closer to 1 as we expect. While the classical method produces similar PWC between E and other meteorological factors in most cases especially for the coherence between E and T after excluding the effects of others (Fig. S6 of Sect. S3 in the Supplement), large differences between these two methods can also be observed. For example, while the new method recognizes the strong coherence between E and RH after excluding the effect of T at scales of around 1 year (Fig. 3d), this coherence was negligible by the classical method (Fig. 5a). Mean PWC values by the new method were consistently higher than the classical method, and the differences ranged from 0.4 to 0.6 around the scale of 1 year (Fig. 5b). Considering the real coherence (Eq.15) rather than complex coherence (Eq.14) between every two variables in the numerators can potentially result in large underestimation of the partial wavelet coherence. Therefore, the ability of the new method to produce more accurate results than the classical method is one of its advantages.

[Figure]

Figure 5.

Partial wavelet coherency (PWC) between evaporation (E) and relative humidity (RH) after excluding the effect of mean temperature (T) using the classical method (Eq. 15) (a) and differences in PWC between the new method (Eq.14) and classical method as a function of scale (b)."

***Comment #3:***
*(3) Regarding the structure, is it more suitable to reorganize the Section 3 and 4, that is, the artificial data and their results are analyzed and discussed in Section 3, while those of real data are analyzed and discussed in Section 4?*

***Response #3:***

Thanks for the good suggestion on paper structure. In the revision, we followed the order of data description, data analysis, results and discussion for each of artificial dataset and real data. To reduce the length of this paper, we have taken the suggestion from Referee #2 to remove the real data related to soil water content by adding more about the introduction of the wavelet methods and in-depth discussion of the advantages and weaknesses of the new method.

Thanks again for your constructive comment.

**Response to Anonymous Referee #2**

*Comment #1:*

***Summary*** *In this technical note, the authors propose a method for identifying relationships between two variables for the case where the two variables are correlated to other variables themselves. They apply their updated partial wavelet coherency' (PWC) method to a synthetic dataset and two real-world applications and show that this updated PWC model shows similar performance as existing PWC models. They conclude that their model outperforms existing models because it provides phase information and allows for excluding several correlated variables from the PWC.*

*Response #1:*

Many thanks for your comment. We think the new method outperforms the existing one from the three aspects: (1) more accurate results because of the theoretical differences (as explained in the **Response #2** to **Referee #1** above); (2) inclusion of phase information; and (3) any number of excluding variables can be considered.

Below we will respond to each of your comments.

*Comment #2:*

***General remarks*** *I think that the study addresses a question of interest to the hydrological community, i.e. 'how can we identify the most important driving variables of a certain phenomena at different time scales'. The technical note is generally well structured. However, I think that it lacks a didactical and detailed introduction to the topic, problem, and wavelet analysis. The introduction would significantly benefit from providing examples of when the identification of bivariate relationships are important (i.e. providing a motivation for the study), an in-depth introduction to wavelet analysis (for the readers who are not yet too familiar with the topic), and an introduction to the terminology used. Extending the introduction will increase the length of the note and I suggest removing the practical example number 2 instead. I think it does not provide additional insights regarding the performance of the method proposed compared to the statements*

*that were already made based on the synthetic data and the first practical example. Since the new method does not seem to clearly outperform existing methods, I would better explain why adding phase information and excluding several confounding variables is beneficial for the analysis. I would also add a more detailed discussion of model weaknesses, especially the implications of detecting spurious correlations. In addition, the note would profit from careful language editing.*

**Response #2:**

More detailed information on the general wavelet analysis, PWC, and problem of existing methods were added in the Introduction and Theory sections (see more details below).

The importance of bivariate relationships was explained at Lines 48-57.

An introduction to wavelet analysis in general was added at Lines 41-43.

The original motivation to have both real datasets is to demonstrate that the proposed method can be used for both spatial and temporal data. We agree that more detailed introduction will increase the length of the paper, so we removed the results related to soil water content dataset.

The differences between the new method and the existing method have been explained in the **Response #2** to the **Referee #1** above.

A separate discussion section was added by including a more detailed discussion of model advantages (e.g., the three aspects mentioned in the Response #1) and weaknesses (including spurious correlations and multiple-testing). Please refer to the discussion section at Lines 399-486.

Language has been carefully checked by editors from our publication office.

Please see the details below on how we will address the comments you have made.

**Major points**

**Comment #3:**
*1. **Abstract:** The abstract is not very accessible to non-wavelet-specialists. I would provide a short example for when such an analysis would be necessary/beneficial and shortly summarize what wavelet coherence analysis is all about. Please also shortly explain why PWC has been introduced in the first place (l. 12). I would also mention the datasets used for model evaluations (l. 14). I think the statement 'producing more accurate results' (l. 18) needs justification, otherwise it is not very credible. I would exclude lines 21-24 because this is a technical note and specific results regarding the example applications going beyond model performance are in my opinion not of interest here.*

**Response #3:**

We have added "Bivariate wavelet coherency is a measure of correlation between two spatial (or time) series in the location-scale (or time-frequency) domain. It is particularly suited to geoscience where relationships between multiple variables commonly differ with locations or/and scales because of various processes involved." to explain what is wavelet coherence and when it would benefit (Lines 9-12).

The PWC was introduced "to detect the scale-specific and localized bivariate relationships by excluding the effects of other variables". (Lines 14-15).

The description of dataset used for model evaluations is "Both stationary and non-stationary artificial datasets with the response variable being the sum of five cosine waves at 256 locations are used to test the methods." (Lines 18-19).

Why the new method produces more accurate results was explained by adding "Compared with the previous PWC calculation, the new method produces more accurate results where there is one excluding variable. This is because bivariate real coherence rather than the bivariate complex coherence was mistakenly used in the previous PWC calculation, which underestimates the PWC" (Lines 22-25).
Lines 21-24 from the previous submission have been removed.

***Comment #4:***
***2. Introduction:*** *The introduction should in my opinion provide a motivation for the use of PWC methods, also for non-specialists on the topic e.g. by providing examples of important bivariate relationships in the geosciences and why we may be interested in them. In addition, an introduction to wavelet analysis in general and wavelet coherence analysis in particular should be provided. The reader should also be made familiar with the terminology used, e.g. what kind of scales are you talking about and what is an 'excluding variable'. A clear motivation for why excluding variables and including phases matters is required to underline the benefits of the methods later on in the results and conclusions sections (l. 57-58). Currently, the introduction does not very well prepare readers for what they are going to read in the methods and results sections.*

***Response #4:***

The importance of bivariate relationships was explained by adding "The BWC partitions correlation between two variables into different locations and scales, which are different from the overall relationships at the sampling scale as shown by the traditional correlation coefficient. For example, BWC analysis indicated that soil water content of a hummocky landscape in the Canadian Prairies was negatively correlated to soil organic carbon content at a slope scale (50 m), but they were positively correlated at a watershed scale (120 m) in summer because of the different processes involved at different scales (Hu et al., 2017). Because the positive correlation may cancel out with the negative at different scales and/or locations, the traditional correlation coefficient between soil water content and soil organic carbon content does not differ significantly from zero, which is misleading." (Lines 48-57).

The motivation for the use of PWC method is further explained by adding "Partial correlation analysis is one such method to avoid the misleading relationships resulting from the interdependence between other variables and both predictor and response variables (Kenney and Keeping, 1939)" (Lines 68-70) and "For example, PWC analysis indicated that Southern Oscillation Index and Pacific Decadal Oscillation did not affect precipitation across India, while this was misinterpreted by the BWC analysis because of their interdependence on Niño 3.4 that affects precipitation (Rathinasamy et al., 2017)" (Lines 78-81).

An introduction to wavelet analysis in general was added as "Wavelet analyses are based on wavelet transform using mother wavelet function which expands spatial (or time) series into location-scale (or time-frequency) space for identification of localized intermittent scales (or frequencies)." (Lines 41-43).

When we talk about scale, it can mean spatial or temporal scale depending on if the dataset are spatial series or time series. To avoid repeatedly addressing if this is related to spatial or time scale, we has defined it at the first time by adding "For convenience, we will mainly refer to location and scale irrespective of spatial or time series unless otherwise mentioned.". (Lines 43-45).

Excluding variable refers to "variable that influences the response variable is excluded". (Lines 82-83).

The explanation on the motivation for why excluding variables and including phases matter was added as "The coherence between response and predictor variables can still be misleading if more than one variable is interdependent with the predictor variable. This is especially true if these variables are correlated with the predictor variable at different locations and/or scales. In addition, without phase information, it is hard to tell if the correlation at a location and scale is positive or negative" in the introduction at Lines 84-88.

**Comment #5:**

**3. Theory:** *I think that you should start even simpler here and provide a short introduction to wavelet analysis (difference between discrete and complex, terminology) and wavelet coherence analysis. In addition, it is unclear to me what exactly the difference between classical PWC and your proposed method is (l. 74-76). Currently, it is not entirely clear to me how the Monte Carlo experiment was performed (l. 108-110). Could you please slightly expand this section?*

**Response #5:**

We has added the introduction to wavelet analysis, wavelet coherence analysis and associated equations at start of the Theory section. Here we assume you mean difference between discrete wavelet transform and continuous wavelet transform. These were added at Lines 101-120 as follows:

"Wavelet analysis is based on the calculations of wavelet coefficients using wavelet transform at different locations and scales for each variable involved. Two types of wavelet transform exist including continuous wavelet transform and discrete wavelet transform. While the discrete wavelet transform is mainly used for data compression and noise reduction, the continuous wavelet transform is widely used for extracting scale-specific and localized features, as is the case of this study (Grinsted et al., 2004). For the continuous wavelet transform, the Morlet wavelet is used as a mother wavelet function to transform a spatial (or time) series into location-scale (or time-frequency) domain, which allows us to identify both location-specific amplitude and phase information of wavelet coefficients at different scales (Torrence and Compo, 1998). From wavelet coefficients, auto- and cross-wavelet power spectra for two variables can be calculated as the product of wavelet coefficient and the complex conjugate of itself (auto-wavelet power spectra) or another variable (cross-wavelet power spectra). The BWC is calculated as the ratio of smoothed cross-wavelet power spectra of two variables to the product of their auto-wavelet power spectra (Grinsted et al., 2004). Hu and Si (2016) extended wavelet coherence from two to multiple ($\geq$3) variables and developed MWC. Detailed information on the calculations of wavelet coefficients, auto- and cross-wavelet power spectra, BWC, and MWC based on the continuous wavelet transform can be found elsewhere (Torrence and Compo, 1998; Grinsted et al., 2004; Si and Farrell, 2004; Si, 2008; Hu and Si, 2016; Hu et al., 2017). Here, we will only introduce the theory and calculation that is very relevant to the PWC. "

In addition, the derivation of Eq.(1) in the original submission from equations of complex partial spectrum in frequency domain and bivariate complex coherence from time-frequency domain was added in the supplement as below:

" **S1 Derivation of the complex PWC Eq.(1)**

Complex partial spectrum from frequency (scale)domain (Makhtar et al., 2014) can be used to define that of time-frequency (location-scale) domain, $\underset{W}{\leftrightarrow}^{y,x\cdot Z}(s,\tau)$, which is expressed as

$$\underset{W}{\leftrightarrow}^{y,x\cdot Z}(s,\tau) = \underset{W}{\leftrightarrow}^{y,x}(s,\tau) - \frac{\underset{W}{\leftrightarrow}^{y,Z}(s,\tau)\overline{\underset{W}{\leftrightarrow}^{x,Z}(s,\tau)}}{\underset{W}{\leftrightarrow}^{Z,Z}(s,\tau)} \tag{S1}$$

where $\underset{W}{\leftrightarrow}$ is the smoothed cross spectrum, $\overline{(\cdot)}$ is the complex conjugate operator, $y$, $x$, and $Z$ ($Z = \{Z_1, Z_2, \cdots, Z_q\}$) refer to the response variable, predictor variable, and excluding variables, respectively. $s$ and $\tau$ refer to scale (frequency) and location (time), respectively.

Given the definition of coherence between two variables $y$ and $x$, their complex coherence $\gamma_{y,x}(s,\tau)$ (Eq.(5)) can be re-written as

$$\gamma_{y,x}(s,\tau) = \frac{\underset{W}{\leftrightarrow}^{y,x}(s,\tau)}{\sqrt{\underset{W}{\leftrightarrow}^{y,y}(s,\tau)\underset{W}{\leftrightarrow}^{x,x}(s,\tau)}} \tag{S2}$$

Then we can define complex partial coherence as

$$\gamma_{y,x\cdot Z}(s,\tau) = \frac{\underset{W}{\leftrightarrow}^{y,x\cdot Z}(s,\tau)}{\sqrt{\underset{W}{\leftrightarrow}^{y,y\cdot Z}(s,\tau)\underset{W}{\leftrightarrow}^{x,x\cdot Z}(s,\tau)}} \tag{S3}$$

Based on Eq. (S1) and Eqs 2, 3, and 4 ($R^2_{y,x,Z}(s,\tau) = \frac{\underset{W}{\leftrightarrow}^{y,Z}(s,\tau)\underset{W}{\leftrightarrow}^{Z,Z}(s,\tau)^{-1}\overline{\underset{W}{\leftrightarrow}^{x,Z}(s,\tau)}}{\underset{W}{\leftrightarrow}^{y,x}(s,\tau)}$,

$R^2_{y,Z}(s,\tau) = \frac{\underset{W}{\leftrightarrow}^{y,Z}(s,\tau)\underset{W}{\leftrightarrow}^{Z,Z}(s,\tau)^{-1}\overline{\underset{W}{\leftrightarrow}^{y,Z}(s,\tau)}}{\underset{W}{\leftrightarrow}^{y,y}(s,\tau)}$, and $R^2_{x,Z}(s,\tau) = \frac{\underset{W}{\leftrightarrow}^{x,Z}(s,\tau)\underset{W}{\leftrightarrow}^{Z,Z}(s,\tau)^{-1}\overline{\underset{W}{\leftrightarrow}^{x,Z}(s,\tau)}}{\underset{W}{\leftrightarrow}^{x,x}(s,\tau)}$ )

we obtain

$$\underset{W}{\leftrightarrow}^{y,x\cdot Z}(s,\tau) = \underset{W}{\leftrightarrow}^{y,x}(s,\tau)\left(1 - \frac{\underset{W}{\leftrightarrow}^{y,Z}(s,\tau)\overline{\underset{W}{\leftrightarrow}^{x,Z}(s,\tau)}}{\underset{W}{\leftrightarrow}^{Z,Z}(s,\tau)\underset{W}{\leftrightarrow}^{y,x}(s,\tau)}\right) = \underset{W}{\leftrightarrow}^{y,x}(s,\tau)\left(1 - R^2_{y,x,Z}(s,\tau)\right) \tag{S4}$$

$$\underset{W}{\leftrightarrow}^{y,y\cdot Z}(s,\tau) = \underset{W}{\leftrightarrow}^{y,y}(s,\tau)\left(1 - \frac{\underset{W}{\leftrightarrow}^{y,Z}(s,\tau)\overline{\underset{W}{\leftrightarrow}^{y,Z}(s,\tau)}}{\underset{W}{\leftrightarrow}^{Z,Z}(s,\tau)\underset{W}{\leftrightarrow}^{y,y}(s,\tau)}\right) = \underset{W}{\leftrightarrow}^{y,y}(s,\tau)\left(1 - R^2_{y,Z}(s,\tau)\right) \tag{S5}$$

$$\underset{W}{\leftrightarrow}^{x,x\cdot Z}(s,\tau) = \underset{W}{\leftrightarrow}^{x,x}(s,\tau)\left(1 - \frac{\underset{W}{\leftrightarrow}^{x,Z}(s,\tau)\overline{\underset{W}{\leftrightarrow}^{x,Z}(s,\tau)}}{\underset{W}{\leftrightarrow}^{Z,Z}(s,\tau)\underset{W}{\leftrightarrow}^{x,x}(s,\tau)}\right) = \underset{W}{\leftrightarrow}^{x,x}(s,\tau)\left(1 - R^2_{x,Z}(s,\tau)\right) \tag{S6}$$

Inserting Eqs S4, S5, and S6 into Eq. (S3), we have

$$\gamma_{y,x\cdot Z}(s,\tau) = \frac{\underset{W}{\leftrightarrow}^{y,x}(s,\tau)\left(1-R^2_{y,x,Z}(s,\tau)\right)}{\sqrt{\underset{W}{\leftrightarrow}^{y,y}(s,\tau)\left(1-R^2_{y,Z}(s,\tau)\right)\underset{W}{\leftrightarrow}^{x,x}(s,\tau)\left(1-R^2_{x,Z}(s,\tau)\right)}} = \frac{\underset{W}{\leftrightarrow}^{y,x}(s,\tau)\left(1-R^2_{y,x,Z}(s,\tau)\right)}{\sqrt{\underset{W}{\leftrightarrow}^{y,y}(s,\tau)\underset{W}{\leftrightarrow}^{x,x}(s,\tau)}\sqrt{\left(1-R^2_{y,Z}(s,\tau)\right)\left(1-R^2_{x,Z}(s,\tau)\right)}} =$$

$$\frac{\left(1-R^2_{y,x,Z}(s,\tau)\right)\gamma_{y,x}(s,\tau)}{\sqrt{\left(1-R^2_{y,Z}(s,\tau)\right)\left(1-R^2_{x,Z}(s,\tau)\right)}} \tag{S7}$$

Obviously, Eq. (S7) and Eq. (1) are identical."

The differences between the new method and the existing method in case of one excluding variable have been explained in the **Response #2** to the **Referee #1** above. By comparing Eq. (14) (new method) and (15) (classical method) , we can conclude that theoretically the classical method underestimates PWC relative to the new one.

Monte Carlo method was explained in more details by adding why we chose AR1 model and how many repeats are needed as we explained in the **Response #1** to the **Referee #1** above.

***Comment #6:***
***4. Data and analysis:*** *I would recommend removing the 'soil water content' example (section 4.2.2) because as I can see it does not show anything that has not yet been shown by the 'free evaporation example' in terms of the validity of the model. I would rather invest the space in extending the introduction as outlined in more details above. In the figure captions, I would add a reference to the dataset used to generate it. In addition, I am not sure what you would like to show with the cases where the variable of interest is excluded. I would therefore exclude the results referring to this exercise (e.g. Figure 1 last row and see l. 236-237). I also think that the figures would profit a lot from using labels for subfigures, which would facilitate orientation. To me, the difference between the Mihanovic et al. (2009) model and the proposed model are not evident by looking at the Figures presented (a difference of 0.03 does not seem to be a lot, l.293). Therefore, I think the actual advantages of using this new method should better be worked out and explained before a statement such as 'the new method outperforms the Mihanovic et al. method' (l. 293-294) is made. Please also explain why the inclusion of 'phase information' is an advantage of the new method (l. 312-313).*

***Response #6:***
Thanks for this advice. We have removed the soil water content example.

Reference to the dataset used to generate the figures was added in the figure caption as "All variables were generated by following Yan and Gao (2007) and Hu and Si (2016) and explained in Section 3.1 and are shown in Fig. S2 of Sect. S3 in the Supplement."

The purpose of showing the cases of variable of interest being excluded is to basically show that the PWC values should be theoretically zero in that case. As we have the similar results in the case of two excluding variables (Figure 3 in the original submission and Figure 2 in the current version), we have removed this from Figure 1.

We have added a label for each subfigure in the revision.

 As we explained above, theoretical differences exist between these two methods in case of one excluding variable. This has been discussed at Lines 185-204.

In the new discussion section, we have highlighted the advantages and weakness of the new method at Lines 399-486 (Please see the details in the **Response #7** below).

***Comment #7:***

***5. A proper discussion section is missing:*** *I would add an in-depth discussion of the weaknesses and benefits of the approach and put the new method into perspective by comparing it to existing methods.*

*Response #7:*
Advantages and weaknesses of the method were added in the discussion section as:

[revised manuscript text omitted]

**Comment #8:**

**6. Conclusions:** *Given the evidence provided in the results section, statements such as 'the new method produces slightly more accurate coherence' do not seem to be justified. As mentioned earlier the benefits of including phase information and excluding several variables need to be better explained. Some of the material presented in this section could be moved to the new discussion section.*

**Response #8:**
As we replied above, we think 'the new method produces more accurate coherence' is justified by considering both the theoretical differences and the example of real data (Figure 5) explained above. The benefits of including phase information and excluding several variables were discussed in the new discussion section as we explained in the Response #7.
Yes, a large part from the conclusions part was moved to the Discussion section as shown in Response #7.

**Comment #9:**

**7. Code availability:** *I would provide the Matlab code via a data/file repository such as HydroShare or Zenodo instead of the supplement (l.27). This would be very helpful for the community and potential users.*

**Response #9:**
We have provided the Matlab code to the figshare (https://figshare.com/s/bc97956f43fe5734c784). Meanwhile, we have also put the updated codes for multiple wavelet coherence (MWC) which is necessary for calculating PWC in the same repository. We have improved the calculation time for MWC.

**Minor points**

***Comment #10:***
*L. 31: please explain what you mean by 'time and space localization'.*
***Response #10:***
We have added an example to show the localization "For example, time series of air temperature usually fluctuates periodically at different scales (e.g., daily and yearly), but abrupt changes in air temperature (e.g., extremely high or low) may occur at certain time points as a result of extreme weather and climate events (e.g., heat and rain)." (Lines 35-38).

***Comment #11:***

*L.34: 'among these methods'*

*Transition from l. 42 to l. 43: very sharp transition from bivariate relationships to prediction. I would try to establish a clear link between the two things.*

***Response #11:***
We have changed "Among which" to "Among these wavelet methods". (Line 45).

We're sorry that we are not sure we understood this comment. But we end up with the wide application of multiple wavelet coherence (MWC) method in the previous graph, and the next paragraph we start with what the MWC application has told us. Namely more predictor variables does not necessarily explain more variations in the response variable because predictor variables are usually cross-correlated. Because of the same reason, bivariate relationships can be misleading. Then we call the need to develop partial wavelet coherence (PWC). Now in the revision, we have put them in the same paragraph.

***Comment #12:***

*L. 48: what do you mean by 'this issue'?*

***Response #12:***
We mean "the misleading relationships resulting from the interdependence between other variables and both predictor and response variables". (Lines 68-70).

***Comment #13:***
*L. 50: what kind of scales? Temporal or spatial?*

***Response #13:***
We mean either temporal or spatial scales depending on if the dataset are time series or spatial series. For avoiding repeatedly saying this, we has clarified this at the first time by adding "For convenience, we will mainly refer to location and scale irrespective of spatial or time series unless otherwise mentioned". (Lines 43-45).

***Comment #14:***
*L. 53-54: would combine greenhouse gas emissions and climate in one category.*

***Response #14:***
Actually we mean different things. We mean precipitation by climate, so we changed climate to meteorology for avoiding confusing.

***Comment #15:***
*L. 61: information 'which will allow to….'*

***Response #15:***
We changed the whole sentence to "this paper aims to develop a PWC method that considers more than one excluding variable and presents phase information. This method reveals the magnitude and type of bivariate relationships after removing the effects from all potentially interdependent variables." at Lines 89-92.

***Comment #16:***
*L. 61: what do you mean by 'analogy' in this context. I think that rephrasing may be required.*

***Response #16:***
We have changed "in analogy with" simply to "from".

***Comment #17:***
*L. 62: Be specific with what you mean by 'it': 'the proposed method'.*

***Response #17:***
We have changed it to "The proposed method".

***Comment #18:***
*L. 76: Please explain to the reader what you mean by 'scale' and 'location'.*

***Response #18:***
Scale and location for spatial series correspond to frequency (periodicity) and time, respectively. As mentioned above, we have added "For convenience, we will mainly refer to location and scale irrespective of spatial or time series unless otherwise mentioned". (Lines 43-45).

***Comment #19:***
*L. 99: same for 'phase angle'.*

***Response #19:***
We have added its explanation in the bracket as "(i.e., angle between two complex numbers)" at Line 153.

***Comment #20:***
*L. 184-185: can in my opinion be removed.*

***Response #20:***
We have removed this sentence.

***Comment #21:***
*L. 191: what does data refer to? Soil water content?*

***Response #21:***
It refers to soil water datasets. Now removed as you suggested.

***Comment #22:***
*L. 214: 'significance band'.*

***Response #22:***
We have changed it to significance band.

*Comment #23:*
*L. 215-216: is this statement underlined by any analysis performed?*

*Response #23:*
Yes. The number is obtained from calculation.

*Comment #24:*
*L. 247: what is the purpose of replacing half of the time series by 0?*

*Response #24:*
As we highlighted in Section 3.1, "second half of the original series of y2 (or z2) are replaced by 0 to simulate abrupt changes (i.e., transient and localized feature) of the spatial series". (Lines 227-228).

*Comment #25:*
*L. 261-263: Which feature in the plots actually indicates these 'abrupt changes'?*

*Response #25:*
The abrupt changes were captured by the abrupt transition from coherence of 0 to coherence of 1 as shown in figure 1i and 1m of current version (top 2 at the left hand side of figure 2 in the original submission).

*Comment #26:*
*L. 266: I can only see one wavelet band of high significance in Figure 3. Where is the second one you mention here?*

*Response #26:*
We did not show the results here, but it was shown in Fig. 2 of our previous paper (Hu and Si, 2016). For this reason, the citation of "(Hu and Si, 2016)" was added here.

*Comment #27:*
*L. 298: introduce term 'octave'.*

*Response #27:*
We have added the explanation "octave refers to the scaled distance between two scales with one scale being twice or half of the other." (Lines 466-467).

*Comment #28:*
*L. 363-366: would move this sentence to discussion section.*

*Response #28:*
Yes, we have moved this sentence to the discussion section.

Thanks again for your constructive comment.

**Response to Anonymous Referee #3**

Anonymous Referee #3

*Comment #1:*

*In this paper, the authors presented an improved variant of PWC for identifying the relationship between variables. This should be reflected in the title (like Improved PWC etc to be included in the title) to convey novel contribution. Also at present it is misleading like the authors proposes PWC concept.*

*Response #1:*

Many thanks for your comments. We have changed the title to "Technical Note: Improved partial wavelet coherency for understanding scale-specific and localized bivariate relationships in geosciences".

*Overall the paper is well written. I recommend for minor revision.*

***Comment #2:***

*Line 18– and producing more accurate results.- pl give quantitative statements*

***Response #2:***
As the two methods in case of one excluding variables have theoretical differences, the outperformance is obvious. However, the degree of outperformance depends, in the case of our artificial dataset, the new method produces PWC values more close to 1 than the existing method as we expect although the difference is not big (e.g., PWC value of 1.0 versus 0.97 between $y$ and $y_2$ at the scale of 8 after excluding the effect of $y_4$). However, the comparison of these two methods using real data indicated that the difference between the two methods can be large. For example, the differences in PWC between evaporation (E) and relative humidity (RH) after excluding the effect of mean temperature (T) can be 0.4-0.6 at the scales of about 1 year. For this reason, rather than giving quantitative statements, we have pointed out why the proposed method produces more accurate results by changing the sentence to "Compared with the previous PWC calculation, the new method produces more accurate results where there is one excluding variable. This is because bivariate real coherence rather than the bivariate complex coherence was mistakenly used in the previous PWC calculation, which underestimates the PWC.". (Lines 22-25).

***Comment #3:***
*Line 31- provide the developments in chronological order – should be checked at all places*
*What is the real advantage in bringing the phase information in practical cases? this should be mentioned in the introduction section*

***Response #3:***
All citations were changed in a chronological order.
The importance of phase information have been explained by adding "without phase information, it is hard to tell if the correlation at a location and scale is positive or negative." (Lines 87-88)

***Comment #4:***

*Line 109 .. sufficient number of times using : : :pl make it clear*
***Response #4:***

Discussion on the sufficient number of times was added as we explained in the **Response #1** to the **Referee #1** above.

***Comment #5:***
*Line 214- significance band*
***Response #5:***

We have changed it to significance band.

***Comment #6:***
*Conclusion: Avoid the statements like – 'this new method produces slightly more accurate coherence'*

***Response #6:***

We have changed it to "Compared with the previous PWC method, the new PWC method has the advantage of dealing with more than one excluding variable and providing the phase information (i.e., correlation type) associated with the PWC. In the case of one excluding variable, this new method produces more accurate coherence than the previous PWC method because the former considers complex coherence between every two variables, while the latter only considers the real coherence "(Lines 492-497).

***Comment #7:***

*Line 450-455 should be explained better ; how can you overcome such problems ? I think better to provide a discussion section before conclusion where such*
*references and unfamiliar terms can be explained in a better way. Then conclusion*
*section should be presented as more specific*

***Response #7:***

New discussion section was be added by moving this part to the discussion section. In terms of spurious correlations and multiple-testing problem, we have put it to a new section 5.2 weaknesses. Meanwhile, the advantages was mentioned in section 5.1. Please see the detailed revision at Lines 399-486 which has also shown above.

Thanks again for your constructive comment.

[revised manuscript text omitted]
 developed in analogy withfrom the partial coherency in the multiple multi-variate spectral partial coherency in easethe frequency (scale) domain (Koopmans, 1995). ItThe proposed method is first tested with artificial datasets following Yan and Gao (2007) and Hu and Si (2016) to demonstrate its capability of capturing the known relationships of the artificial data. Next, the new method is compared with the Mihanović et al. (2009) method. Then it is applied to two a real (i.e., field measured) dataset. i.e.,s in geosciences including temporal time series of free water evaporation at the Changwu site in China (Hu and Si, 2016) and spatial series of soil water content from a transect in the hummocky landscape in Saskatchewan, Canada (Biswas and Si, 2011a; Hu et al., 2017). These two datasets are chosen because the MWC results previously presented (Hu and Si, 2016) can be used to assess the new method.Finally, the advantages and weaknesses of the new method are discussed by comparing it with the previous PWC method.

**2. Theory**

Wavelet analysis is based on the calculations of wavelet coefficients using wavelet transform at different locations and scales for each variable involved. Two types of wavelet transform exist including continuous wavelet transform and discrete wavelet transform. While the discrete wavelet transform is mainly used for data compression and noise reduction, the continuous wavelet transform is widely used for extracting scale-specific and localized features, as is the case of this study (Grinsted et al., 2004). For the continuous wavelet transform, the Morlet wavelet is used as a mother wavelet function to transform a spatial (or time) series into location-scale (or time-frequency) domain, which allows us to identify both location-specific amplitude and phase information of wavelet coefficients at different scales (Torrence and Compo, 1998). From wavelet coefficients, auto- and cross- wavelet power spectra for two variables can be calculated as the product of wavelet coefficient and the complex conjugate of itself (auto-wavelet power spectra) or another variable (cross-wavelet power spectra). The BWC is calculated as the ratio of smoothed cross-wavelet power spectra of two variables to the product of their auto-wavelet power spectra (Grinsted et al., 2004). Hu and Si (2016) extended wavelet coherence from two to multiple ($\geq$3) variables and developed MWC. Detailed information on the calculations of wavelet coefficients, auto- and cross-wavelet power spectra, BWC, and MWC based on the continuous wavelet transform can be found elsewhere (Torrence and Compo, 1998;

Grinsted et al., 2004; Si and Farrell, 2004; Si, 2008; Hu and Si, 2016; Hu et al., 2017). Here, we will only introduce the theory and calculation that is very relevant to the PWC.

Similar to BWC and MWC, PWC is calculated from auto- and cross-wavelet power spectra, for the response variable $y$, predictor variable $x$, and excluding variables $Z$ ($Z =$

$\{Z_1, Z_2, \cdots, Z_q\}$). Koopmans (1995) developed the multivariate complex PWC in the frequency (scale) domainIn analogy with the partial coherency in the multivariate spectral case (Koopmans, 1995),. Here, we extend the Koopmans (1995) method from the frequency (scale) domain to the time-frequency (location-scale) domain. Therefore, the complex PWC

between $y$ and $x$ after excluding variables $Z$ at scale $s$ and location $\tau$ ,

$\gamma_{y,x\cdot Z}(s,\tau)$ , can be written as

$\gamma_{y,x\cdot Z}(s,\tau)$

$$= \frac{\left(1 - R^2_{y,x\cdot,Z}(s,\tau)\right)  \gamma_{y,x}(s,\tau)}{ \sqrt{\left(1 - R^2_{y,Z}(s,\tau)\right)\left(1 - R^2_{x,Z}(s,\tau)\right)}} \tag{1}$$

where $R^2_{y,x\cdot,Z}(s,\tau)$, $R^2_{y,Z}(s,\tau)$, and $R^2_{x,Z}(s,\tau)$ can be calculated by following Hu and Si (2016) as

$$R^2_{y,x\cdot,Z}(s,\tau) = \frac{\overset{\leftrightarrow^{y,Z}}{W}(s,\tau) \, \overset{\leftrightarrow^{Z,Z}}{W}(s,\tau)^{-1} \, \overline{\overset{\leftrightarrow^{x,Z}}{W}(s,\tau)}}{\overset{\leftrightarrow^{y,x}}{W}(s,\tau)} \tag{2}$$

$$R^2_{y,Z}(s,\tau) = \frac{\overset{\leftrightarrow^{y,Z}}{W}(s,\tau) \, \overset{\leftrightarrow^{Z,Z}}{W}(s,\tau)^{-1} \, \overline{\overset{\leftrightarrow^{y,Z}}{W}(s,\tau)}}{\overset{\leftrightarrow^{y,y}}{W}(s,\tau)} \tag{3}$$

$$R^2_{x,Z}(s,\tau) = \frac{\overset{\leftrightarrow^{x,Z}}{W}(s,\tau) \, \overset{\leftrightarrow^{Z,Z}}{W}(s,\tau)^{-1} \, \overline{\overset{\leftrightarrow^{x,Z}}{W}(s,\tau)}}{\overset{\leftrightarrow^{x,x}}{W}(s,\tau)} \tag{4}$$

Eq. (1) can be also derived analogously from the complex partial spectrum for the frequency domain and the definition of complex coherence between two variables in the time- frequency domain (see the Supplement (Sect. S1) for the derivation process). Note that

$R^2_{y,x\cdot Z}(s,\tau)$ is a matrix with complex values while $R^2_{y,Z}(s,\tau)$ and $R^2_{x,Z}(s,\tau)$ are matrices with real numbers.

$\gamma_{y,x}(s,\tau)$  is the complex wavelet coherence between $y$ and $x$, which can be written as

$\gamma_{y,x}(s,\tau)$  $= \dfrac{\overset{y,x(s,\tau)}{\overleftrightarrow{W}}}{\left(\overset{y,y(s,\tau)}{\overleftrightarrow{W}}\overset{x,x(s,\tau)}{\overleftrightarrow{W}}\right)^{1/2}}$ (5)

where $\underset{(\cdot)}{\leftrightarrow}$ is the smoothing operator, $\overline{(\cdot)}$ is the complex conjugate operator, $(\cdot)^{-1}$

indicates the inverse of the matrix, and

$\overset{y,Z}{\underset{W}{\leftrightarrow}}(s,\tau) = \left[\overset{y,Z_1}{\underset{W}{\leftrightarrow}}(s,\tau) \ \overset{y,Z_2}{\underset{W}{\leftrightarrow}}(s,\tau) \cdots \overset{y,Z_q}{\underset{W}{\leftrightarrow}}(s,\tau)\right]$ (6)

$\overset{x,Z}{\underset{W}{\leftrightarrow}}(s,\tau) = \left[\overset{x,Z_1}{\underset{W}{\leftrightarrow}}(s,\tau) \ \overset{x,Z_2}{\underset{W}{\leftrightarrow}}(s,\tau) \cdots \overset{x,Z_q}{\underset{W}{\leftrightarrow}}(s,\tau)\right]$ (7)

$\overset{Z,Z}{\underset{W}{\leftrightarrow}}(s,\tau) = \begin{bmatrix} \overset{Z_1,Z_1}{\underset{W}{\leftrightarrow}}(s,\tau) & \cdots & \overset{Z_1,Z_q}{\underset{W}{\leftrightarrow}}(s,\tau) \\ \vdots & \ddots & \vdots \\ \overset{Z_q,Z_1}{\underset{W}{\leftrightarrow}}(s,\tau) & \cdots & \overset{Z_q,Z_q}{\underset{W}{\leftrightarrow}}(s,\tau) \end{bmatrix}$ (8)

where $\overset{A,B}{\underset{W}{\leftrightarrow}}(s,\tau)$ is the smoothed auto-wavelet power spectra (when $A=B$) or cross- wavelet power spectra (when $A\neq B$) at scale $s$ and location $\tau$, respectively.

The squared PWC (hereinafter referred to as PWC) at scale $s$ and location $\tau$, $\rho^2_{y,x\cdot Z}$, can be written as

$\rho^2_{y,x\cdot Z} = \dfrac{\left|1-R^2_{y,x\cdot Z}(s,\tau)\right|^2 R^2_{y,x}(s,\tau)}{\left(1-R^2_{y,Z}(s,\tau)\right)\left(1-R^2_{x,Z}(s,\tau)\right)}$ (9)

where $R^2_{y,x}(s,\tau)$ is squared BWC between $y$ and $x$, which can be expressed as

$R^2_{y,x}(s,\tau) = \dfrac{\overset{y,x(s,\tau)}{\overleftrightarrow{W}}\overline{\overset{y,x(s,\tau)}{\overleftrightarrow{W}}}}{\overset{y,y(s,\tau)}{\overleftrightarrow{W}}\overset{x,x(s,\tau)}{\overleftrightarrow{W}}}$ (10)

[revised manuscript text omitted]

---

## Referee Report (RR1)

**Review for manuscript "Technical note: Improved partial wavelet coherency for understanding scale-specific and localized bivariate relationships in the geosciences"**

**Authors:** Wei Hu and Bing Si

**Journal:** Hydrology and Earth System Sciences Discussions

**General remarks**

I think that the authors tried to include most of my previous comments. The introduction is now a bit more accessible even though I personally would probably even provide more background for the non-expert reader. The paper refocused on one instead of two practical examples which reduces its length and it provides a new discussion section that discusses both the weaknesses and advantages of the new method. Overall, the motivation and benefits of the new method seem much clearer now. However, I still think that the note would profit substantially from careful language editing and from a few clarifications now and then. The line numbers I use in my more detailed comments below refer to the 'track-changes' version of the revised document.

**Major points**

- It is not entirely clear to me what you mean by 'spatial series' (e.g. l. 11-12 and many other instances in the text). Do you mean to refer to a 'spatial field' or to 'spatio-temporal' data sets? Please clarify.
- Please pay attention to the use of articles. They seem to be missing in some places (l.43 'as the spatial distribution', l.53 'on the wavelet transform using a mother…') and can be removed in others (e.g. l. 16. 'detect relationships' instead of 'detect the relationships', l. 38 'untangle scale-specific').
- I understand now that you are trying to demonstrate that using the Mihanovic PWC with a complex instead of a real-valued component is crucial. I think that you should/could be even clearer about that in the introduction. I think line 110 would be a great spot to talk about the deficits of previous implementation of PWC (see l. 219-225). I.e. make it clear that PWC has been proposed by Mihanovic and used by others in a wrong way. What you are proposing is a correct interpretation rather than a new method. Did I understand this correctly?
- 79-83: Here, it might be good to provide an example for what such 'other variables' could be and why their influence can blur the relationship between a response and predictor variable.
- L. 128: I would keep the description of the complex wavelet transform a bit more general and mention that different types of mother wavelets can be used among which one is the Morlet wavelet. Also consider mentioning the properties of the Morlet wavelet that make it particularly suitable for the application in PWC.
- L. 199: I do not agree that 'AR(1) can be used to simulate most geoscience data very well'. Indeed, many hydrological time series show long-range dependencies, which are not captured by AR(1)s. What does this mean for your Monte Carlo experiment? Should it be rerun using a more appropriate dependence structure? Maybe, this is also just something for the discussion section where you may want to discuss what type of autocorrelation structures other may want to use of AR(1) if long-range dependence was an issue.

- In Section 3.1, you introduce variables y1 to y5 and z1 to z5. Subsequently, you only seem to use y2, y4, z2 and z4. Why is it necessary to introduce all of them if just some of them are used? Seemed confusing to me. Could you just remove all the other (unnecessary) variables?
- L.261-272 talks about the case where one variable is excluded and l.273-280 about the case when two variables are excluded. This could be made clearer by starting the paragraphs e.g. with First,… Second,…
- L. 273-280: I guess I do not fully understand what you are trying to say in that paragraph. What I understand is that you are saying that excluding one or several variables does not make a difference, i.e. it is sufficient to exclude one variable. If so, why is the proposed method necessary given that one of its biggest advantages is that it can exclude several variables? Please clarify.

**Minor points**

- In the abstract, the reader does not yet know what 'the previous PWC calculation' is (l. 30), which means that some alternative phrasing is needed there.
- L. 57: what do 'these wavelet methods' refer to? Please specify.
- L. 67: 'the negative one'
- L. 69: 'can be misleading.
- L.106: 'provides phase information'
- L. 108: 'an extension of'
- L. 133: what does 'itself' refer to?
- L. 139: instead of 'elsewhere' I would write 'e.g. in …'
- What do the different R terms refer to if you had to describe that in one summary sentence?
- L. 196: what does 'sufficient' mean in terms of the number of iterations?
- L.283-284: rephrasing needed
- Figure 1 and others: I would add some labels for the stationary and non-stationary case. The arrows mentioned seem really tiny and are hardly visible.
- L. 601-603: indicate that your method corrects for this. Furthermore, there is a problem with the brackets.
- L. 675: what do you mean by 'multiple-testing problem'?
- L. 723: rephrasing needed
- L. 726: 'analyze' instead of 'detect'

---

## Author Response (AR2)

**Response to Editor Dr. Bettina Schaefli**

Comments to the Author:
Dear Authors the paper was re-reviewed by one of the initial reviewers and I think he/she has valid points for further improvement. The question whether the introduction should give more details for non expert reader is a question of taste. I understand your viewpoint that the paper is a technical note addressed more towards advanced users.

**Response:**
Thanks for giving us another chance to further improve the paper. We tried to give more explanations to make the paper more accessible, although we agree that the paper may be of more interest to readers who have basic knowledge on wavelet analysis.
Below are the detailed explanations on how we revised the paper.

**Response to Anonymous Referee #2**
Comments from Referee #2

**Review for manuscript "Technical note: Improved partial wavelet coherency for understanding scale-specific and localized bivariate relationships in the geosciences"**
**Authors:** Wei Hu and Bing Si
**Journal:** Hydrology and Earth System Sciences Discussions

*Comment #1:*
**General remarks**
I think that the authors tried to include most of my previous comments. The introduction is now a bit more accessible even though I personally would probably even provide more background for the non-expert reader. The paper refocused on one instead of two practical examples which reduces its length and it provides a new discussion section that discusses both the weaknesses and advantages of the new method. Overall, the motivation and benefits of the new method seem much clearer now. However, I still think that the note would profit substantially from careful language editing and from a few clarifications now and then. The line numbers I use in my more detailed comments below refer to the 'track-changes' version of the revised document.

*Response #1:*
Thanks for your constructive comments again. We have tried our best to address all the concerns you raised and made further clarifications in places where we see necessary. We have asked an Editor from the Science Publication Office of our institute to edit the language.
Please see the detailed explanations below on how we revised this manuscript.

**Major points**

*Comment #2:*
It is not entirely clear to me what you mean by 'spatial series' (e.g. l. 11-12 and many other instances in the text). Do you mean to refer to a 'spatial field' or to 'spatio-temporal' data sets? Please clarify.

*Response #2:*
Yes, we mean spatial data collected from spatial field by "spatial series". Basically we want to use spatial data and time series to distinguish data from spatial domain and time domain. As spatial data is widely known as geospatial data, we changed spatial series to spatial data. So, this sentence was changed to "Bivariate wavelet coherency is a measure of correlation between two variables in the location-scale (spatial data) or time-frequency (time series) domain".

**Comment #3:**
Please pay attention to the use of articles. They seem to be missing in some places (l.43 'as the spatial distribution', l.53 'on the wavelet transform using a mother…') and can be removed in others (e.g. l. 16. 'detect relationships' instead of 'detect the relationships', l. 38 'untangle scale-specific').

**Response #3:**
Thanks. We have corrected the inappropriate use of articles. In addition, one Editor from our Science Publication Office has checked the language including the use of articles for us.

**Comment #4:**
I understand now that you are trying to demonstrate that using the Mihanovic PWC with a complex instead of a real-valued component is crucial. I think that you should/could be even clearer about that in the introduction. I think line 110 would be a great spot to talk about the deficits of previous implementation of PWC (see l. 219-225). I.e. make it clear that PWC has been proposed by Mihanovic and used by others in a wrong way. What you are proposing is a correct interpretation rather than a new method. Did I understand this correctly?

**Response #4:**
First, we think it's useful to add the deficit of previous PWC calculation when we point out the research gap in the previous paragraph. So we add "Unfortunately, the PWC calculation in many previous studies (Ng and Chan, 2012b; Rathinasamy et al., 2017; Aloui et al., 2018; Altarturi et al., 2018b; Jia et al., 2018; Li et al., 2018; Mutascu and Sokic, 2020; Wu et al., 2020) was based on an incorrect Matlab code developed by Ng and Chan (2012a) who might have misinterpreted the equation of Mihanović et al. (2009) and mistakenly used bivariate real coherence rather than bivariate complex coherence for calculating PWC." at Lines 90-95.

In the place you suggested, we add "We expect that the new method produces more accurate PWC values than the calculation of Ng and Chan (2012a) where there is one excluding variable." (Lines 106-108) to illustrate that one of the aims is to improve the PWC calculation in the case of one excluding variable.

To clarify, our method is a new method in terms of (1) dealing with multiple excluding variables and (2) providing phase information. This is our motivation and has been pointed out at Lines 95-102. On the other hand, in the case of one excluding variable, new method improved the calculation of partial wavelet coherence.

**Comment #5:**
79-83: Here, it might be good to provide an example for what such 'other variables' could be and why their influence can blur the relationship between a response and predictor variable.

**Response #5:**
An example was given by adding the following text to explain how other variables can blur the relationship between a response and predictor variable: "For example, soil water content of the root zone was found to be positively related to grass yield throughout the year in a small watershed on the Chinese Loess Plateau (Hu et al., 2017a). This was because higher grass yield usually coincided with finer soils that usually have higher water holding capacity. After removing the effects of other factors including sand content, partial correlation analysis indicated that soil water content was negatively affected by grass yield during growing seasons and not affected by grass yield during non-growing seasons as expected. The study of Hu et al. (2017a) clearly demonstrated that partial correlation analysis can be an effective method to avoid misleading relationships between response (e.g., soil water content) and predictor variables (e.g., grass yield) when the latter was interdependent with other variables (e.g., sand content)." (Lines 68-78)

***Comment #6:***
L. 128: I would keep the description of the complex wavelet transform a bit more general and mention that different types of mother wavelets can be used among which one is the Morlet wavelet. Also consider mentioning the properties of the Morlet wavelet that make it particularly suitable for the application in PWC.

***Response #6:***
We have revised it as "Wavelet analysis is based on the wavelet transform, which includes continuous wavelet transform and discrete wavelet transform. While the discrete wavelet transform is mainly used for data compression and noise reduction, the continuous wavelet transform is widely used for extracting scale-specific and localized features, as in the case of this study (Grinsted et al., 2004). The wavelet transform decomposes the spatial (or time series) data into a set of location- and scale-specific wavelet coefficients, which are scaled (contracted or expanded) and shifted versions of mother wavelets. Different mother wavelets are available for wavelet transform. Among which, the Morlet wavelet, composed of a complex exponential multiplied by a Gaussian window, provides a good balance between location and scale localization. Therefore, continuous wavelet transform with the Morlet wavelet is suitable to transform spatial (or time series) data into a location-scale (or time-frequency) domain,…". (Lines 117-128)

We kept the description of two types of wavelet transform (e.g., continuous wavelet transform and discrete wavelet transform) because we think it would be useful to highlight that the continuous wavelet transform is widely used for extracting scale-specific and localized features, as is the case of this study.

***Comment #7:***

L. 199: I do not agree that 'AR(1) can be used to simulate most geoscience data very well'. Indeed, many hydrological time series show long-range dependencies, which are not captured by AR(1)s. What does this mean for your Monte Carlo experiment? Should it be rerun using a more appropriate dependence structure? Maybe, this is also just something for the discussion section where you may want to discuss what type of autocorrelation structures other may want to use of AR(1) if long-range dependence was an issue.

***Response #7:***
We agree with you. For this reason, we changed the sentence to "The first-order autoregressive model (AR(1)) is chosen because most geoscience data can be effectively simulated by it (Wendroth et al., 1992; Grinsted et al., 2004; Si and Farrell, 2004), although we recognize that time series with long-range dependence is also common in many areas such as hydrology (Szolgayová et al., 2014)" (Lines 188-192)

In the case of long-range dependence data, then high-order autoregressive models may be used to generate noise series for significance test. This was discussed as one weaknesses in the discussion section 5.2 as "The AR(1) model was used to generate noise series for testing the confidence level of PWC. High-order autoregressive models rather than AR(1) may be beneficial for a significance test where spatial (or time series) data are characterized by long-range dependence (Szolgayová et al., 2014)." (Lines 508-511)

*Comment #8:*
In Section 3.1, you introduce variables y1 to y5 and z1 to z5. Subsequently, you only seem to use y2, y4, z2 and z4. Why is it necessary to introduce all of them if just some of them are used? Seemed confusing to me. Could you just remove all the other (unnecessary) variables?

*Response #8:*
We think it's useful to mention y1 to y5 and z1 to z5 as the response variable y and z are the sum of these five cosine waves. We will not have good understanding of the response variable y and z without the characteristics of these five cosine functions. However, we changed the description of predictor and excluding variables at Lines 241-243 as "The predictor and excluding variables (Fig. S1 of Sect. S4 in the Supplement) are selected from two of the five cosine waves (i.e., $y_2$ and $y_4$ or $z_2$ and $z_4$) and/or their derivatives" Hope this will not be confusing anymore.

*Comment #9:*
L.261-272 talks about the case where one variable is excluded and l.273-280 about the case when two variables are excluded. This could be made clearer by starting the paragraphs e.g. with First,... Second,...

*Response #9:*
Done, thanks. Please see the changes at Lines 245 and 259.

*Comment #10:*
L. 273-280: I guess I do not fully understand what you are trying to say in that paragraph. What I understand is that you are saying that excluding one or several variables does not make a difference, i.e. it is sufficient to exclude one variable. If so, why is the proposed method necessary given that one of its biggest advantages is that it can exclude several variables? Please clarify.

*Response #10:*
Sorry for the confusion here. We did not mean that results of excluding one variable are the same to those of excluding several variables.
In the case of one excluding variable, we calculated the PWC between response variable (y or z) and predictor variable y2 or z2 (results are presented), as well as predictor variable y4 or z4 (results are not presented).

Because the results in case of predictor variable of y4 (z4) are analogous to those in case of predictor variable of $y_2$ (or $z_2$), we chose not to show the results for the case of predictor variable of $y_4$ (or $z_4$).

So the sentences "Note that PWC between y (or z) and other predictor variables (e.g., $y_4$ or $z_4$) after excluding $y_2$ or $z_2$ and their equivalent derivative variables (i.e., noised variables or variables with 0) are also calculated. The related results are not shown because they are analogous to those in case of predictor variable of $y_2$ (or $z_2$)." should have been placed in the end of previous paragraph that introduces the case of one excluding variable.

 For avoiding confusion, we simply removed these sentences during the revision.

**Minor points**
*Comment #11:*

In the abstract, the reader does not yet know what 'the previous PWC calculation' is (l. 30), which means that some alternative phrasing is needed there.

*Response #11:*
We have changed this sentence to "Where there is one excluding variable, the new method produces higher and more accurate PWC values than the previous PWC calculation that mistakenly used bivariate real coherence rather than bivariate complex coherence in the calculation."

*Comment #12:*
L. 57: what do 'these wavelet methods' refer to? Please specify.

*Response #12:*
We refer to all wavelet methods that are based on wavelet transform. For avoid confusion, we removed "Among these wavelet methods" as this does not affect our understanding of the coming sentence.

*Comment #13:*
L. 67: 'the negative one'

*Response #13:*
Done. Thanks.

*Comment #14:*
L. 69: 'can be misleading.

*Response #14:*
Done. Thanks.

*Comment #15:*
L.106: 'provides phase information'

*Response #15:*
Done. Thanks.

*Comment #16:*
L. 108: 'an extension of'

*Response #16:*
Done. Thanks.

*Comment #17:*
L. 133: what does 'itself' refer to?

*Response #17:*
"itself" in previous copy refer to the variable for calculating auto-wavelet power spectra which is the product of wavelet coefficient and its complex conjugate.

In the revision, we introduced generally the calculation of auto-wavelet power spectra and cross-wavelet power spectra by changing it simply to: "Wavelet coefficients and their complex conjugates are used to calculate auto-wavelet power spectra and cross-wavelet power spectra". (Lines 129-131)

*Comment #18:*
L. 139: instead of 'elsewhere' I would write 'e.g. in …'

*Response #18:*
To avoid long sentence by listing many citations in the format of author (year), we changed elsewhere to "in previous studies", and put all citations in the brackets after "e.g.,".

*Comment #19:*
 What do the different R terms refer to if you had to describe that in one summary sentence?

*Response #19:*
R refers to bivariate wavelet coherence (in case of two variables) or multiple wavelet coherence (in case of more than two variables).

*Comment #20:*
L. 196: what does 'sufficient' mean in terms of the number of iterations?

*Response #20:*
We mean the minimum number of iterations required that produces small error (e.g., relative difference <2%) in mean PWC. So, we add "minimum number required" in the brackets after "sufficient number".

*Comment #21:*
L.283-284: rephrasing needed

*Response #21:*
We changed it to "Theoretically, we expect (a) PWC is 1 at scales corresponding to relative complement of excluding variable scales in predictor variable scales, and 0 at other scales." based on the set theory.

*Comment #22:*
Figure 1 and others: I would add some labels for the stationary and non-stationary case. The arrows mentioned seem really tiny and are hardly visible.

*Response #22:*
We have added labels "Stationary" and "Non-stationary" at the right hand side of each row. The blurry arrows are partly related to the PDF conversion. We have made the arrows sparser and bigger. Although the revised one may not look perfect, this should not be a problem when the original copy of the figure (.tif format) is used for final publication.

*Comment #23:*
L. 601-603: indicate that your method corrects for this. Furthermore, there is a problem with the brackets.

*Response #23:*
Done.

***Comment #24:***
L. 675: what do you mean by 'multiple-testing problem'?

***Response #24:***
It means that "more than one individual hypothesis is tested simultaneously". We have added the explanation.

***Comment #25:***
L. 723: rephrasing needed

***Response #25:***
We have changed it to "Application of the new method to the real dataset has further proved its robustness in untangling the bivariate relationships after removing the effects of all other variables in multiple location-scale domains"( Lines 522-524)

***Comment #26:***
 L. 726: 'analyze' instead of 'detect'

***Response #26***
Done.
Thanks again for your constructive comments.

[revised manuscript text omitted]

---

## Author Response (AR3)

**Response to Editor Dr. Bettina Schaefli**

Comments to the Author:
Dear Authors
thanks for the additional detailed revisions. The paper is now ready for publication.

**Response:**
*Thanks for your further comments. We have addressed all and the changes are explained below. We have also carefully read through the paper and made some minor changes in terms of wording.*

Non-public comments to the Author:
I have some final technical comments:

Abstract: I am not sure it is possible to have the link to the code in the abstract, ask the editorial team for advice.

**Response:**
*We have emailed the editorial team and asked for their advice. Unfortunately, we have not heard from them yet. But we have checked the guide for authors, and it seems there are no limits on the use of link in the abstract. Our purpose to have the link to the code in the abstract is to give readers a fast way to access the codes if they are interested in this method without reading the whole paper. In order not to miss the deadline for returning the revised copy and also the reasons we explain here, we decide to keep the link at this stage.*

The start of the conclusion section is not entirely clear; the sentence "Partial wavelet coherency (PWC) is developed in this study" reads like if this was the first time that this is proposed.

**Response:**
*We changed it to "Partial wavelet coherency (PWC) is improved …".*

Furthermore, the conclusion suggests that the previously suggested PWC method was wrong whereas only the implementation was wrong.

Current text: In the case of one excluding variable, this new method produces more accurate coherence than the previous PWC method calculation that considered only real coherence rather than complex coherence between every two variables.

Suggestion: In the case of one excluding variable, the PWC implementation provided here (in the paper and the published code) produces more accurate coherence than the previously published PWC implementation that considered wrongly real coherence rather than complex coherence between every two variables

**Response:**
*We changed it as you suggested. Similar changes were also made for the sentence in the abstract. Thanks.*

Comments on the math notation:

• There are several instances left where there is dot before the capital Z in the subscript

**Response:**
*The dot is the notation for excluding variables, so we explained it as "where symbol · is the notation for excluding variables," to avoid confusion.*

• Z1 has the 1 not in the subscript, I hope that the typesetting can correct this

**Response:**
*We've put number 1 into the subscript position for all in the main paper and Supplement. Thanks.*

• HESS does not accept /like multi-letter variable names; y2wn and similar are not well chosen, I suggest to use y subscript 2,w, or 2,m, 2,s or 2,h0

**Response:**
*We have done as you suggest. For example, they are read as something like $y_{2,h0}$, $y_{2,w}$, $y_{2,w,h0}$, and $y_{24}$. Changes are made for both main paper and the Supplement. Thanks.*

[revised manuscript text omitted]

$$\quad R^2_{y,x,Z}(s,\tau) = \frac{\overleftrightarrow{W}^{y,Z}(s,\tau)\,\overleftrightarrow{W}^{Z,Z}(s,\tau)^{-1}\,\overline{\overleftrightarrow{W}^{x,Z}(s,\tau)}}{\overleftrightarrow{W}^{y,x}(s,\tau)} \qquad (2)$$

$$\quad R^2_{y,Z}(s,\tau) = \frac{\overleftrightarrow{W}^{y,Z}(s,\tau)\,\overleftrightarrow{W}^{Z,Z}(s,\tau)^{-1}\,\overline{\overleftrightarrow{W}^{y,Z}(s,\tau)}}{\overleftrightarrow{W}^{y,y}(s,\tau)} \qquad (3)$$

$$\quad R^2_{x,Z}(s,\tau) = \frac{\overleftrightarrow{W}^{x,Z}(s,\tau)\,\overleftrightarrow{W}^{Z,Z}(s,\tau)^{-1}\,\overline{\overleftrightarrow{W}^{x,Z}(s,\tau)}}{\overleftrightarrow{W}^{x,x}(s,\tau)} \qquad (4)$$

Eq. (1) can be also derived analogously from the complex partial spectrum for the frequency domain according to the definition of complex coherence between two variables in the time- frequency domain (see the Supplement (Sect. S1) for the derivation process). Note that

$R^2_{y,x,Z}(s,\tau)$ is a matrix with complex values, while $R^2_{y,Z}(s,\tau)$ and $R^2_{x,Z}(s,\tau)$ are matrices with real numbers. $\gamma_{y,x}(s,\tau)$ is the complex wavelet coherence between $y$ and $x$, which can be written as

$$\quad \gamma_{y,x}(s,\tau) = \frac{\overleftrightarrow{W}^{y,x}(s,\tau)}{\left(\overleftrightarrow{W}^{y,y}(s,\tau)\overleftrightarrow{W}^{x,x}(s,\tau)\right)^{1/2}} \qquad (5)$$

where $\overleftrightarrow{(\cdot)}$ is the smoothing operator, $\overline{(\cdot)}$ is the complex conjugate operator, $(\cdot)^{-1}$

indicates the inverse of the matrix, and

$$\underset{W}{\leftrightarrow}^{y,Z}(s,\tau) = \left[\underset{W}{\leftrightarrow}^{y,Z_1}(s,\tau) \; \underset{W}{\leftrightarrow}^{y,Z_2}(s,\tau) \cdots \underset{W}{\leftrightarrow}^{y,Z_q}(s,\tau)\right] \qquad (6)$$

$$\underset{W}{\leftrightarrow}^{x,Z}(s,\tau) = \left[\underset{W}{\leftrightarrow}^{x,Z_1}(s,\tau) \; \underset{W}{\leftrightarrow}^{x,Z_2}(s,\tau) \cdots \underset{W}{\leftrightarrow}^{x,Z_q}(s,\tau)\right] \qquad (7)$$

$$\underset{W}{\leftrightarrow}^{Z,Z}(s,\tau) = \begin{bmatrix} \underset{W}{\leftrightarrow}^{Z_1,Z_1}(s,\tau) & \cdots & \underset{W}{\leftrightarrow}^{Z_1,Z_q}(s,\tau) \\ \vdots & \ddots & \vdots \\ \underset{W}{\leftrightarrow}^{Z_q,Z_1}(s,\tau) & \cdots & \underset{W}{\leftrightarrow}^{Z_q,Z_q}(s,\tau) \end{bmatrix} \qquad (8)$$

where $\underset{W}{\leftrightarrow}^{A,B}(s,\tau)$ is the smoothed auto-wavelet power spectra (when $A=B$) or cross- wavelet power spectra (when $A\neq B$) at scale $s$ and location , respectively.

The squared PWC (hereinafter referred to as PWC) at scale $s$ and location , $\rho^2_{y,x\cdot Z}$, can be written as

$$\rho^2_{y,x\cdot Z} = \frac{\left|1-R^2_{y,x,Z}(s,\tau)\right|^2 R^2_{y,x}(s,\tau)}{\left(1-R^2_{y,Z}(s,\tau)\right)\left(1-R^2_{x,Z}(s,\tau)\right)} \qquad (9)$$

where $R^2_{y,x}(s,\tau)$ is squared BWC between $y$ and $x$, which can be expressed as

$$R^2_{y,x}(s,\tau) = \frac{\underset{W}{\leftrightarrow}^{y,x}(s,\tau)\overline{\underset{W}{\leftrightarrow}^{y,x}(s,\tau)}}{\underset{W}{\leftrightarrow}^{y,y}(s,\tau)\underset{W}{\leftrightarrow}^{x,x}(s,\tau)} \qquad (10)$$

The phase angle (i.e., angle between two complex numbers) between $y$ and $x$ after excluding effect of $Z$ is

$$\vartheta_{y,x\cdot Z}(s,\tau) = \varphi_{y,x\cdot Z}(s,\tau) + \vartheta_{y,x}(s,\tau) \qquad (11)$$

where

$$\varphi_{y,x\cdot Z}(s,\tau) = \arg\left(1 - R^2_{y,x,Z}(s,\tau)\right) \qquad (12)$$

and $\vartheta_{y,x}(s,\tau)$ is the wavelet phase between $y$ and $x$, which can be expressed as

$$\vartheta_{y,x}(s,\tau) = \tan^{-1}\left(\mathrm{Im}\big(W^{y,x}(s,\tau)\big)/\mathrm{Re}\big(W^{y,x}(s,\tau)\big)\right) \qquad (13)$$

where arg denotes the argument of the complex number, $W^{y,x}(s,\tau)$ is the cross-wavelet power spectrum between $y$ and $x$ at scale $s$ and location $\tau$; Im and Re denote the imaginary and real part of $W^{y,x}(s,\tau)$, respectively.

When only one variable (e.g., $\cancel{Z}Z_1\cancel{1}$) is excluded, Eq.(9) can be written as (see the

Supplement (Sect. S2) for the derivation process)

$$\rho^2_{y,x\cdot Z_1\cancel{1}} = \frac{\left|\gamma_{y,x}(s,\tau)-\gamma_{y,Z_1Z_1}(s,\tau)\overline{\gamma_{x,Z_1Z_1}(s,\tau)}\right|^2}{\left(1-R^2_{y,\cancel{Z}Z_1\cancel{1}}(s,\tau)\right)\left(1-R^2_{x,Z_1\cancel{1}}(s,\tau)\right)} \qquad (14)$$

The widely used Monte Carlo method (Torrence and Compo, 1998; Grinsted et al., 2004;

Si and Farrell, 2004) is used to calculate PWC at the 95% confidence level. In brief, the

PWC calculation is repeated for a sufficient number (i.e., minimum number required) of times using data generated by Monte Carlo simulations based on the first-order autocorrelation coefficient (r1). The first-order autoregressive model (AR(1)) is chosen because most geoscience data can be effectively simulated by it (Wendroth et al., 1992;

Grinsted et al., 2004; Si and Farrell, 2004), although we recognize that time series with long-range dependence is also common in many areas such as hydrology (Szolgayová et al., 2014). Different combinations of r1 values (i.e., 0.0, 0.5, and 0.9) were used to generate

10 to 10 000 AR(1) series with three, four and five variables. Our results indicate that the noise combination has little impact on the PWC values at the 95% confidence level as also found by Grinsted et al. (2004) for the BWC case (data not shown). The relative difference of PWC at the 95% confidence level compared with that calculated from the 10 000 AR(1)

series decreases with the increase in number of AR(1) series (Fig. S1 of Sect. S3 in the

Supplement). When the number of AR(1) is above 300, a very low maximum relative difference (e.g., <2%) is observed. Therefore, a repeating number of 300 seems to be sufficient for a significance test. However, if calculation time is not a barrier, a higher repeating number, such as ≥1000, is recommended. The 95[th] percentile of PWCs of all simulations at each scale represents PWC at the 95% confidence level. The average PWC, percent area of significant coherence (PASC) relative to the whole wavelet location–scale domain (Hu and Si, 2016), and average value of significant PWC (PWC$_{sig}$) are also calculated for different location–scale domains.

In the case of one excluding variable ($Z = \{Z_1\}$), Mihanović et al. (2009) suggested that

PWC can be calculated by an equation analogous to the traditional partial correlation squared (Kenney and Keeping, 1939) without giving detailed derivation process. Their equation is the same as Eq. (14). Unfortunately, Ng and Chan (2012a) might have misinterpreted the equation of Mihanović et al. (2009) and developed Matlab code for calculating PWC using the equation expressed as

$$\rho^2_{y,x \cdot Z_1 Z_1} = \frac{\left| R_{y,x}(s,\tau) - R_{y,Z_1 Z_1}(s,\tau)\, R_{x,Z_1 Z_1}(s,\tau) \right|^2}{\left(1 - R^2_{y,Z_1 Z_1}(s,\tau)\right)\left(1 - R^2_{x,Z_1 Z_1}(s,\tau)\right)} \tag{15}$$

where $R_{y,x}(s,\tau)$, $R_{y,Z_1 Z_1}(s,\tau)$, and $R_{x,Z_1 Z_1}(s,\tau)$ are the square root of $R^2_{y,x}(s,\tau)$,

$R^2_{y,Z_1 Z_1}(s,\tau)$, $R^2_{x,Z_1 Z_1}(s,\tau)$, respectively. $R^2_{y,Z_1 Z_1}(s,\tau)$ and $R^2_{x,Z_1 
[revised manuscript text omitted]
}sn$ (or $z_{2,s}sn$), $y_{2,m}mn$ (or $z_{2,m}mn$), $y_{2,w}wn$ (or $z_{2,w}wn$), $y_{2,h0}h_\theta$ (or $z_{2,h0}h_\theta$), $y_{2,w,h0}wnh_\theta$ (or $z_{2,w,h0}wnh_\theta$), $y_{2,m,h0}mnh_\theta$ (or $z_{2,m,h0}mnh_\theta$), and $y_{2,s,h0}snh_\theta$ (or $z_{2,s,h0}snh_\theta$) for the stationary (or non-stationary) case using the new method. Arrows represent the phase angles of the cross-wavelet power spectra between two variables after eliminating the effect of excluding variables. Arrows pointing to the right (left) indicate positive (negative) correlations. Thin and thick solid lines show the cones of influence and the 95% confidence levels, respectively. All variables were generated by following Yan and Gao (2007) and Hu and Si (2016) and are explained in Section 3.1 and shown in Fig. S2 of Sect. S3 in the Supplement.

Compared with the case of excluding variable of $y_4$ (Fig. 1a), excluding the effect of

$y_{2,s}sn$ (Fig. 1b) results in slightly narrower band of significant PWC and slightly reduced mean $PWC_{sig}$ (0.94 versus 0.96). When less noise is included in the excluding variables (i.e.,

$y_{2,m}mn$ and $y_{2,w}wn$) (Fig. 1c-d), the significant PWC band becomes narrower. The PASC

values are 86%, 77%, and 32% for excluding $y_{2,s}sn$, $y_{2,m}mn$ and $y_{2,w}wn$, respectively, at scales of 6–10. Moreover, the mean $PWC_{sig}$ decreases from 0.94 ($y_{2,s}sn$) to 0.93 ($y_{2,m}mn$)

and 0.89 ($y_{2,w}wn$) when progressively less noise is added (Fig. 1b-d). For the non-stationary case, similar results are obtained (Fig. 1e-h). The only difference is that the scales with significant PWC values change with location, as is found for MWC (Hu and Si, 2016).

When the second half of the excluding variable series is replaced by 0, the PWC values in that half are close to 1, while those in the first half of data series are 0 at scales corresponding to the predictor variable (Fig. 1i and 1m). For the stationary case, after excluding the effect of $y_{2,h0}h_0$, the PWC values are close to 1 (0.98) and 0 in the second and first half of the data series, respectively, at the dimensionless scale of 8 (Fig. 1i). Similar results are observed for the non-stationary case (Fig. 1m). This is anticipated because the series of 0s is independent of the predictor variable and hence has no effect on the correlations between response and predictor variables at these locations. If different magnitudes of noises are added to the first half of the excluding variables ($y_2$ or $z_2$), the significant PWC band in the first half becomes wider as the magnitude of noises increases, while the significant PWC band in the second half remains almost unchanged (Fig. 1j-l and

Fig. 1n-p). In the stationary case, for example, the PASC values at scales of 6–10 are 40%

($y_{2,\text{w,h0}}$), 74% ($y_{2,\text{m,h0}}$), and 86% ($y_{2,\text{s,h0}}$) in the first half, while those values vary from 86% to 90% in the second half (Fig. 1j-l). Meanwhile, the mean PWC$_{\text{sig}}$ in the first half at scales of 6–10 increases from 0.91 to 0.94 in both the stationary (Fig. 1j-l) and non-stationary (Fig. 1n-p) cases as more noises are added to the excluding variable $y_2$ or $z_2$.

This indicates that the new PWC method can also capture the abrupt changes (Fig. 1i and

1m) in the data series, and has the ability to deal with localized relationships.

3.2.2   PWC with two excluding variables using the new method

When both $y_2$ and $y_4$ (or $z_2$ and $z_4$) are considered in the predictor variables, there are two bands of wavelet coherence of 1 between $y$ (or $z$) and $y_{24}$ (or $z_{24}$) (Hu and Si, 2016), which correspond to the scales of two predictor variables. However, after the effect of $y_4$

(or $z_4$) is removed, only one band with PWC of around 1 occurs at the scale of the predictor variable $y_2$ (or $z_2$) (Fig. 2a and 2f). After both predictor variables $y_2$ and $y_4$ (or $z_2$ and $z_4$) are excluded (Fig. 2b and 2g), PWC between $y$ (or $z$) and $y_{24}$ (or $z_{24}$) is 0 at all locationscale domains as expected. When one of the excluding variables $y_2$ (or $z_2$) is added with noises, the relationship between response variable $y$ (or $z$) and predictor variable $y_{24}$ (or

$z_{24}$) becomes significant at scales of the excluding variable $y_2$ (or $z_2$) (Fig. 2c and 2h).

Similar to the case of one excluding variable (Fig. 1), less noise in the excluding variable of $y_2$ (or $z_2$) results in a narrower significant PWC band, and reduced mean PWC$_{\text{sig}}$ values, e.g., from 0.96 ($y_{2,\text{s}}$) to 0.90 ($y_{2,\text{w}}$) in the stationary case (Fig. 2c-e) and from 0.95 ($z_{2,\text{s}}$)

to 0.92 ($z_{2,\text{w}}$) in the non-stationary case (Fig. 2h-j).

[Figure]

**Figure 2.**

Partial wavelet coherency (PWC) between response variable $y$ (or $z$) and predictor variable $y_{24}\cancel{y_4}$ (or $z_{24}\cancel{z_4}$) after excluding the effect of variables $y_4$ (or $z_4$), $y_2+y_4$ (or $z_2+z_4$), $y_{2,s}\cancel{sn}+y_4$ (or $z_{2,s}\cancel{sn}+z_4$), $y_{2,m}\cancel{mn}+y_4$ (or $z_{2,m}\cancel{mn}+z_4$), and $y_{2,w}\cancel{wn}+y_4$ (or $z_{2,w}\cancel{wn}+
[revised manuscript text omitted]